# DualOptim: Enhancing Efficacy and Stability in Machine Unlearning with Dual Optimizers

**Xuyang Zhong**
Department of Computer Science
City University of Hong Kong
xuyang.zhong@my.cityu.edu.hk

**Haochen Luo**
Department of Computer Science
City University of Hong Kong
chester.hc.luo@my.cityu.edu.hk

**Chen Liu** *
Department of Computer Science
City University of Hong Kong
chen.liu@cityu.edu.hk

## Abstract

Existing machine unlearning (MU) approaches exhibit significant sensitivity to hyperparameters, requiring meticulous tuning that limits practical deployment. In this work, we first empirically demonstrate the instability and suboptimal performance of existing popular MU methods when deployed in different scenarios. To address this issue, we propose Dual Optimizer (**DualOptim**), which incorporates adaptive learning rate and decoupled momentum factors. Empirical and theoretical evidence demonstrates that DualOptim contributes to effective and stable unlearning. Through extensive experiments, we show that DualOptim can significantly boost MU efficacy and stability across diverse tasks, including image classification, image generation, and large language models, making it a versatile approach to empower existing MU algorithms. Codes are available at https://github.com/CityU-MLO/DualOptim.

## 1 Introduction

Recent advancements in machine unlearning (MU) research have established it as a crucial technique for utilizing pretrained models while addressing various trustworthy challenges in various applications [1, 2]. These MU methods have two major categories: *exact MU* [3] and *approximate MU* [4]. Exact MU requires retraining a model on a dataset excluding forgetting samples from scratch, which is computationally prohibitive for large-scale models and datasets. Therefore, we study approximate MU methods in this work, with focuses on their efficacy and stability.

Most existing approximate MU methods [5, 6, 7, 8, 9] aim to maximize loss on forget samples while preserving performance on retain samples. While effective, these methods exhibit significant sensitivity to hyperparameter choices, with optimal configurations requiring meticulous tuning that varies substantially across datasets and forgetting scenarios. This dependency complicates their practical deployment. As illustrated in Figure 1, we empirically demonstrate that state-of-the-art MU approaches [6, 7, 8] exhibit suboptimal performance or instability. These limitations highlight the critical need for techniques to improve the effectiveness and stability of MU methods.

In this work, we first reveal the unstable performance of existing MU methods when they are applied in different scenarios. Considering the need to optimize two objectives on the forgetting

---

*Corresponding Author

39th Conference on Neural Information Processing Systems (NeurIPS 2025).

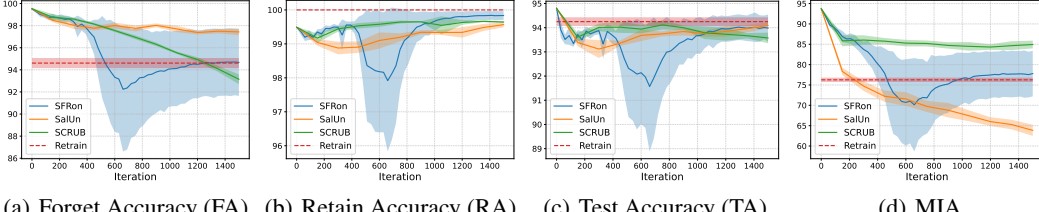

| (a) Forget Accuracy (FA) | (b) Retain Accuracy (RA) | (c) Test Accuracy (TA) | (d) MIA |

Figure 1: Unlearning process of MU baselines. SFRon [7], SalUn [6] and SCRUB [8] are adopted as the baselines. The metrics are those mentioned in Sec. 3.1. All results are obtained from unlearning 10% random subset of CIFAR-10 on ResNet-18. The solid lines and shadows denote the mean and standard deviation across 5 trials with different random data. The hyperparameters of different methods are selected based on minimizing the averaging gap between retraining and them across 5 trials. The red dashed lines denote the final performance of retraining as a reference.

and the retaining samples, we then propose Dual Optimizer (**DualOptim**) to address this challenge. Specifically, we employ an optimizer with adaptive learning rate, such as Adam [10, 11], to optimize the forgetting objective, while utilizing a separate optimizer for the retaining objective. Compared with existing methods, DualOptim decouples the momentum terms for the two objectives by using and updating them separately. In addition, DualOptim is a plug-and-play solution that can be easily integrated into existing MU algorithms.

Our theoretical analysis validates that decoupling momentum terms in two separate optimizers can help decrease the variance of model parameters, indicating a more stable performance. Through comprehensive experiments, we validate the effectiveness of DualOptim in improving the performance of existing MU methods across various tasks, including image classification, image generation, and large language models. Our proposed technique is generic and capable of pushing the state-of-the-art performance on multiple tasks. We summarize the contributions of this paper as follows:

1. We introduce Dual Optimizer (**DualOptim**) to enhance the efficacy and stability of MU methods by incorporating an adaptive learning rate and decoupled momentum. DualOptim can be seamlessly integrated into existing MU algorithms.

2. We provide empirical and theoretical analyses to demonstrate the contribution of DualOptim in improving unlearning performance and stability.

3. Comprehensive experiments across diverse scenarios such as image classification, image generation, and large language models validate the effectiveness of DualOptim in boosting and stabilizing the performance of MU methods.

## 2 Related Works

**Machine Unlearning (MU).** MU targets the need to remove specific data influences from pretrained models [12, 13, 14], while complying with privacy requirements tied to differential privacy [3, 15, 16, 17, 18]. Initially developed on linear classifiers with convex loss objective functions, exact MU [3, 15, 19, 20] allowed precise data removal under privacy budgets to counter privacy attacks. However, exact MU cannot be applied to deep neural networks due to their non-convex loss functions and prohibitive computational cost for full retraining. In this context, approximate MU [4, 21] was proposed to address this issue by fine-tuning the model to achieve the forgetting effect. Despite improved efficiency, approximate MU may cause catastrophic performance declines on retaining data [21, 22]. Recent methods incorporate fine-tuning [6, 8, 23], sparsity regularization [24], knowledge distillation [8, 25], saliency map [6, 7] and alternative updating [7] to better balance efficient forgetting and model utility. However, as illustrated in Figure 1, these methods generally exhibit suboptimal performance or high hyperparameter sensitivity, underscoring the necessity to develop a method to enhance efficacy and stability during unlearning.

**MU for Multiple Tasks.** Besides classification, MU has broad applications for multiple tasks. For example, recent text-conditional *image generation* models have showcased their ability to produce images that closely align with textual descriptions [26, 27, 28, 29]. However, significant security and privacy concerns have been raised [1, 30, 31], necessitating the application of MU methods to these

models. While early works [30, 32, 33, 34] focuses on concept deletion in diffusion models, recent studies [6, 7, 35] improve their performance and propose methods applicable to more general image generators. Another notable example is *large language models (LLMs)*, which have demonstrated remarkable capabilities [36, 37] but also face privacy and copyright issues like retaining unauthorized content [38, 39, 40, 41]. To address these concerns, MU methods have been employed to fine-tune LLMs [2, 9, 42, 43] to effectively and efficiently achieve data forgetting. However, all these methods struggle to achieve a balance between model utility and forget effectiveness. In addition, intensive hyperparameter tuning is expected for optimal performance, bringing challenges for practitioners.

In summary, MU has broad applications across various tasks. However, there are some common issues with existing methods, including high hyper-parameter sensitivity and the utility-forgetting trade-offs. In this work, we propose DualOptim as a generic solution to enhance the efficacy and stability of MU across different tasks.

## 3 Methodology

### 3.1 Preliminary

Let $\mathcal{D} = \{z_i\}_{i=1}^N$ denote the training set for pretraining. The subset of the training set we aim to forget during unlearning is known as the *forget set* $\mathcal{D}_f \subset \mathcal{D}$, and its complement, $\mathcal{D}_r = \mathcal{D} \setminus \mathcal{D}_f$ is the *retain set*. In the context of MU, we denote the parameters of pretrained model as $\theta_o \in \mathbb{R}^d$, which is trained on $\mathcal{D}$. Consistent with previous studies [6, 7, 21, 24], *Retraining* is considered the gold standard for MU, where the model parameters $\theta_r \in \mathbb{R}^d$ are trained from scratch on $\mathcal{D}_r$. However, retraining is computationally intensive or even infeasible, especially for large models and datasets. This poses a significant challenge in the practical applications of MU.

We focus on approximate MU in this work, its primary objective is to obtain an unlearned model, referred to as $\theta_u \in \mathbb{R}^d$, from $\theta_o$ on $\mathcal{D}_f$ and $\mathcal{D}_r$ so that it can serve as an accurate and computationally efficient alternative to the retrained model $\theta_r$. Mathematically, MU aims to solve the optimization problem: $\min_\theta \mathcal{L}_f(\mathcal{D}_f, \theta) + \mathcal{L}_r(\mathcal{D}_r, \theta)$, where $\mathcal{L}_f$ and $\mathcal{L}_r$ denote forget and retain loss objective functions, respectively. In practice, $\mathcal{L}_r$ is usually the same as the loss objective function in pre-training and $\mathcal{L}_f$ is the opposite, so MU aims to improve the performance on $\mathcal{D}_r$ while degrading the performance on $\mathcal{D}_f$. To avoid confusion, we call $\mathcal{L}_f(\mathcal{D}_f, \theta)$, $\mathcal{L}_r(\mathcal{D}_r, \theta)$ that we aim to minimize *forget loss* and *retain loss*, respectively. In addition, many MU algorithms [6, 7, 8] update $\theta$ by alternately optimizing $\mathcal{L}_f(\mathcal{D}_f, \theta)$ and $\mathcal{L}_r(\mathcal{D}_r, \theta)$ for better performance. The generic pipeline of MU is presented in Algorithm 1. For notation simplicity, we use $g_f := \nabla_\theta \mathcal{L}_f(\mathcal{D}_f, \theta)$ and $g_r := \nabla_\theta \mathcal{L}_r(\mathcal{D}_r, \theta)$ to denote the gradients of the forget loss and the retain loss, respectively. In mini-batch updates, we use $\widehat{g}_f$ and $\widehat{g}_r$ to denote the corresponding stochastic gradients.

The numerical analysis in this section is based on classification tasks. We adopt the same evaluation schemes as [7]. That is, we use the accuracy on the forget set (FA), the retain set (RA), test set (TA) and the success rate of membership inference attack (MIA) [44] [2] on the forgetting set to indicate model utility and the forgetting effectiveness. A competitive unlearning method should have stable performance and small gaps with retraining in all these four evaluation schemes.

In the following subsections, we first illustrate the challenges associated with MU and propose corresponding approaches to address them. Ultimately, we introduce **DualOptim**, which integrates these proposed techniques to enhance both efficacy and stability in MU.

### 3.2 Adaptive Learning Rate Enables Stable Forgetting

Despite the effectiveness of existing MU methods, achieving satisfactory unlearning performance often relies on carefully selecting hyperparameters. The optimal hyperparameters can vary significantly across different forget sets, complicating the practical applications of MU algorithms. As shown in Figure 2 (a)-(d), we use the same hyperparameters for SFRon [7], the best-performed baseline in our evaluation, with the default SGD optimizer for 5 trials of different randomly sampled forget sets. We defer more results of other existing methods to Appendix C.2. Despite being intensively tuned, the unified hyperparameters exhibit quite unstable performance across different forget sets during unlearning. This underscores that a hyperparameter configuration effective for certain forget sets may

---

[2]We follow the implementation in [7] and adopt the entropy of the output probabilities as the metric in MIA.

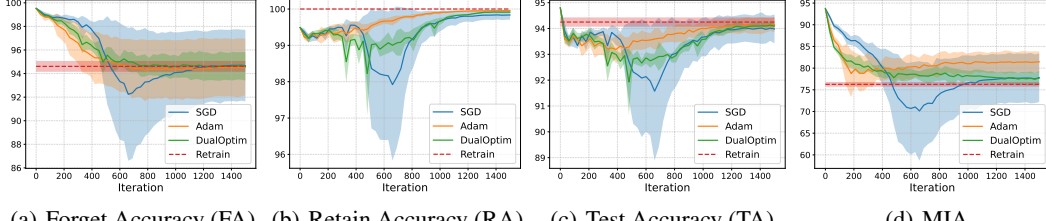

(a) Forget Accuracy (FA)  (b) Retain Accuracy (RA)  (c) Test Accuracy (TA)  (d) MIA

Figure 2: Unlearning process with different ablations of the proposed method. All results are obtained from unlearning 10% random subset of CIFAR-10 by SFRon [7] on ResNet-18. **(a)-(d)** The metrics are those mentioned in Sec. 3.1. The red dashed lines denote the final performance of retraining as a reference. The solid lines and shadows denote the mean and standard deviation across 5 trials with different random forget sets.

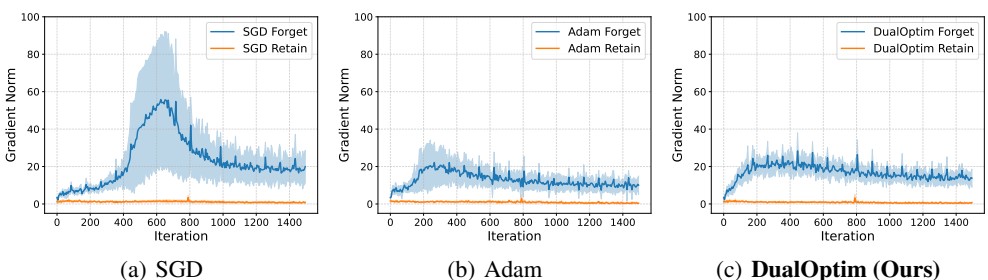

(a) SGD  (b) Adam  (c) **DualOptim (Ours)**

Figure 3: Norms of stochastic forget gradient $\widehat{g}_f$ and stochastic retain gradient $\widehat{g}_r$ using different optimizers. All results are obtained from unlearning 10% random subset of CIFAR-10 by SFRon on ResNet-18. **(a)-(c)** The curves are obtained using SGD, Adam, DualOptim, respectively.

not be suitable for others. When dealing with a new forget set, even if it is sampled in the same way, the need for precise hyperparameter tuning can pose challenges for practitioners.

Figure 3 (a) demonstrates the magnitude of the gradients on the forget set and the retain set when using SGD optimizer. We can clearly see that the gradients on the forget set have significantly larger magnitudes with substantial variations compared with the ones on the retain set. In this context, it is crucial to adaptively adjust the learning rate to handle different gradient magnitudes. Therefore, we use optimizers with preconditioners, such as Adam [10, 11] to evade tricky hyperparameter tuning for the learning rate. Specifically, Adam employs adaptive learning rates based on the historic gradient magnitudes, making it suitable to handle large gradient magnitude variations during unlearning.

The observations in Figure 2 (a)-(d) and Figure 3 (b) suggest that Adam provides more stable performance and induces smaller gradient norm across different forget sets. However, despite enhanced stability, the noticeable performance gap with retraining, especially in the metric of membership inference attack (MIA), indicates that Adam only achieves suboptimal unlearning efficacy. We need a mechanism to improve performance while ensuring the stability of the method.

### 3.3 Decoupled Momentum for Enhanced Stability in Machine Unlearning

Inspired by the disparities in the magnitudes of $\widehat{g}_f$ and $\widehat{g}_r$, along with their non-positive correlation observed in Figure 3 and 4, we introduce *decoupled momentum*, which employs two separate momentum terms dedicated to the forget loss $\mathcal{L}_f$ and the retain loss $\mathcal{L}_r$, respectively. This approach ensures that the momentum update for the forget loss remains unaffected by the gradients of the retaining loss, and vice versa. By decoupling the momentum terms in this manner, we can more effectively optimize the distinct processes of forgetting and retaining, leading to enhanced stability and performance. As illustrated in Figure 2 (a)-(d) (represented by the green lines) and Figure 3 (c), the combination of adaptive learning rate with decoupled momentum not only stabilizes the unlearning process but also leads to improved unlearning efficacy. Note that as presented in Table 5 and Table 6 of Appendix C.2, although other methods like SalUn [6] and SCRUB [8] perform more stable than SFRon, they only achieve suboptimal unlearning performance. Nevertheless, our method can also improve their performance.

Besides empirical findings, we provide a theoretical analysis to elucidate how decoupled momentum contributes to stability. Before the detailed analyses, we first establish the following assumptions.

**Assumption 3.1. (Stochastic Gradient Condition)** For all time steps $t = 0, \ldots, T-1$, the stochastic gradients of the forget loss $\widehat{\boldsymbol{g}}_{f,t}$ and retain loss $\widehat{\boldsymbol{g}}_{r,t}$ satisfy:

$$\widehat{\boldsymbol{g}}_{f,t} = \boldsymbol{g}_{f,t} + \boldsymbol{\epsilon}_{f,t}, \quad \widehat{\boldsymbol{g}}_{r,t} = \boldsymbol{g}_{r,t} + \boldsymbol{\epsilon}_{r,t}, \tag{1}$$

where $\boldsymbol{g}_{f,t} := \nabla_{\theta_t} \mathcal{L}_f(\mathcal{D}_f, \theta_t)$ and $\boldsymbol{g}_{r,t} := \nabla_{\theta_t} \mathcal{L}_r(\mathcal{D}_r, \theta_t)$ are the full-batch gradients with model parameter $\theta_t$ at the time stamp $t$. $\boldsymbol{\epsilon}_{f,t}$ and $\boldsymbol{\epsilon}_{r,t}$ are batch noises with zero mean and a bounded variance: there exists a minimal $\sigma^2 \geq 0$ such that $\mathrm{Var}(\boldsymbol{\epsilon}_{f,t}) \leq \sigma^2$, $\mathrm{Var}(\boldsymbol{\epsilon}_{r,t}) \leq \sigma^2$ for all $t$.

**Assumption 3.2. (Correlation Bounds)** The correlation between the stochastic gradients from the same function in different time steps is bounded while the correlation between stochastic gradients from different functions is non-positive. That is to say, $\exists \tau \in [0,1]$ such that:

$$\forall t_1 \neq t_2, , s.t. \ \rho(\widehat{\boldsymbol{g}}_{f,t_1}, \widehat{\boldsymbol{g}}_{f,t_2}) \leq \tau, \rho(\widehat{\boldsymbol{g}}_{r,t_1}, \widehat{\boldsymbol{g}}_{r,t_2}) \leq \tau, \quad \forall t_1, t_2, \rho(\widehat{\boldsymbol{g}}_{f,t_1}, \widehat{\boldsymbol{g}}_{r,t_2}) \leq o(\tau) \simeq 0 \tag{2}$$

The assumption $\forall t_1, t_2, \rho(\widehat{\boldsymbol{g}}_{f,t_1}, \widehat{\boldsymbol{g}}_{r,t_2}) \leq o(\tau) \simeq 0$ is motivated by the observations in Figure 4 of Appendix C.1 and for sake of notation simplicity. Our analyses can be easily extended to the cases if we use a small constant to bound this correlation.

**Assumption 3.3. (Lipschitz Smoothness)** The loss functions $\mathcal{L}_f$ and $\mathcal{L}_r$ are both $L$-smooth:

$$\forall \theta_1, \theta_2, \|\nabla_{\theta_1} \mathcal{L}_f(\mathcal{D}_f, \theta_1) - \nabla_{\theta_2} \mathcal{L}_f(\mathcal{D}_f, \theta_2)\| \leq L\|\theta_1 - \theta_2\|, \tag{3}$$

$$\forall \theta_1, \theta_2, \|\nabla_{\theta_1} \mathcal{L}_r(\mathcal{D}_r, \theta_1) - \nabla_{\theta_2} \mathcal{L}_r(\mathcal{D}_r, \theta_2)\| \leq L\|\theta_1 - \theta_2\|. \tag{4}$$

We assume the same Lipschitz constant for $\mathcal{L}_f$ and $\mathcal{L}_r$ because (1) $\mathcal{D}_f$ and $\mathcal{D}_r$ are from similar distributions; (2) $L_f$ and $L_r$ are usually opposite functions.

We now consider the SGD update scheme with a shared or decoupled momentum factor. Since we alternately update the parameters based on the stochastic gradients from $\mathcal{L}_f$ and $\mathcal{L}_r$, we use $\{(\boldsymbol{m}_{f,t}^S, \boldsymbol{m}_{r,t}^S)\}_{t=0}^{T-1}$ to denote the momentum factors after using the gradients from the forget loss and the retain loss, respectively, for time stamp $t$ when we are using the shared momentum. Similarly, we use $\{(\boldsymbol{m}_{f,t}^D, \boldsymbol{m}_{r,t}^D)\}_{t=0}^{T-1}$ to denote the momentum factors when using the decoupled momentum. We use $\{(\theta_{f,t}^S, \theta_{r,t}^S)\}_{t=0}^{T-1}$, $\{(\theta_{f,t}^D, \theta_{r,t}^D)\}_{t=0}^{T-1}$ to represent the corresponding updated parameters. We use $\alpha \in [0,1]$ to denote the momentum factor and $\eta$ as the learning rate, then the update schemes for a shared and decoupled momentum factors are shown as follows:

$$
\begin{aligned}
\text{(Shared Momentum)} &\begin{cases} \boldsymbol{m}_{f,t}^S &= \alpha \boldsymbol{m}_{r,t-1}^S + \widehat{\boldsymbol{g}}_{f,t}^S, \quad \theta_{f,t}^S = \theta_{r,t-1}^S - \eta \boldsymbol{m}_{f,t}^S \\ \boldsymbol{m}_{r,t}^S &= \alpha \boldsymbol{m}_{f,t}^S + \widehat{\boldsymbol{g}}_{r,t}^S, \quad \theta_{r,t}^S = \theta_{f,t}^S - \eta \boldsymbol{m}_{r,t}^S \end{cases} \\
\text{(Decoupled Momentum)} &\begin{cases} \boldsymbol{m}_{f,t}^D &= \alpha \boldsymbol{m}_{f,t-1}^D + \widehat{\boldsymbol{g}}_{f,t}^D, \quad \theta_{f,t}^D = \theta_{r,t-1}^D - \eta \boldsymbol{m}_{f,t}^D \\ \boldsymbol{m}_{r,t}^D &= \alpha \boldsymbol{m}_{r,t-1}^D + \widehat{\boldsymbol{g}}_{r,t}^D, \quad \theta_{r,t}^D = \theta_{f,t}^D - \eta \boldsymbol{m}_{r,t}^D \end{cases}
\end{aligned} \tag{5}
$$

Here, we use $\{(\widehat{\boldsymbol{g}}_{f,i}^S, \widehat{\boldsymbol{g}}_{r,i}^S)\}_{i=1}^{T-1}$ and $\{(\widehat{\boldsymbol{g}}_{f,i}^D, \widehat{\boldsymbol{g}}_{r,i}^D)\}_{i=1}^{T-1}$ to denote the stochastic gradients during unlearning in the case of shared momentum and the decoupled momentum, respectively. Based on Algorithm 1, we update the parameters by gradients from the forget loss and the retain loss alternately. Therefore, we have $\boldsymbol{g}_{f,i}^S = \nabla_\theta \mathcal{L}_f(\theta_{r,i-1}^S)$, $\boldsymbol{g}_{f,i}^D = \nabla_\theta \mathcal{L}_f(\theta_{r,i-1}^D)$, $\boldsymbol{g}_{r,i}^S = \nabla_\theta \mathcal{L}_r(\theta_{f,i}^S)$, $\boldsymbol{g}_{r,i}^D = \nabla_\theta \mathcal{L}_r(\theta_{f,i}^D)$, and $\widehat{\boldsymbol{g}}_{f,i}^S, \widehat{\boldsymbol{g}}_{f,i}^D, \widehat{\boldsymbol{g}}_{r,i}^S, \widehat{\boldsymbol{g}}_{r,i}^D$ are their stochastic variants.

When using decoupled momentum factors, the momentum factors $\{\boldsymbol{m}_{f,0}^D, \boldsymbol{m}_{f,1}^D, ..., \boldsymbol{m}_{f,T-1}^D\}$, $\{\boldsymbol{m}_{r,0}^D, \boldsymbol{m}_{r,1}^D, ..., \boldsymbol{m}_{r,T-1}^D\}$ are two independent sequences. By contrast, when using a shared momentum factor, the factor is updated as a sequence $\{\boldsymbol{m}_{f,0}^S, \boldsymbol{m}_{r,0}^S, \boldsymbol{m}_{f,1}^S, \boldsymbol{m}_{r,1}^S, ..., \boldsymbol{m}_{f,T-1}^S, \boldsymbol{m}_{r,T-1}^S\}$. When initialization, we let $\theta_{r,-1}^S = \theta_{r,-1}^D = \theta_o$ and $\boldsymbol{m}_{r,-1}^S = \boldsymbol{m}_{f,-1}^D = \boldsymbol{m}_{r,-1}^D = 0$.

We focus on the variance of the parameters in two different update schemes in (5), both of which iteratively calculate the gradients on random variables. Therefore, we first estimate the variance caused by gradient calculation as in the following lemma.

**Lemma 3.4. (Variance of Gradients)** *If the loss function $\mathcal{L}$ is Lipschitz smooth with a constant $L$, and $\mathrm{Var}(\theta) \leq \sigma_\theta^2$, then we have $\mathrm{Var}(\nabla_\theta \mathcal{L}(\theta)) \leq L^2 \sigma_\theta^2$.*

The proof and discussions are deferred to Appendix A.1. We can use Lemma 3.4 to derive the maximum variance of model parameters for both update schemes in (5) as in following theorem.

**Theorem 3.5.** *(**Variance Bound Comparison for Decoupled vs. Shared Momentum**) For the update schemes indicated in (5) using the same hyperparameters ($\eta$, $\alpha$), and we use $\overline{\mathrm{Var}}(\cdot)$ to denote the maximum variance of a variable, if the function $\mathcal{L}_f$, $\mathcal{L}_r$ and the stochastic gradient $\{(\widehat{\boldsymbol{g}}_{f,i}^S, \widehat{\boldsymbol{g}}_{r,i}^S)\}_{i=0}^{T-1}$, $\{(\widehat{\boldsymbol{g}}_{f,i}^D, \widehat{\boldsymbol{g}}_{r,i}^D)\}_{i=0}^{T-1}$ satisfy Assumption 3.1, 3.2, and 3.3, then*

$$\forall t, \overline{\mathrm{Var}}(\theta_{f,t}^D) \leq \overline{\mathrm{Var}}(\theta_{f,t}^S), \quad \overline{\mathrm{Var}}(\theta_{r,t}^D) \leq \overline{\mathrm{Var}}(\theta_{r,t}^S), \tag{6}$$

The proof is deferred to Appendix A.2. Theorem 3.5 formalizes that decoupled momentum induces smaller worst-case parameter variance compared to shared momentum, demonstrating that decoupled momentum theoretically enhances the stability of unlearning. Additionally, we prove that decoupled momentum can also induce smaller worst-case variance of downstream metrics in Appendix A.3.

## 3.4 Dual Optimizers for Machine Unlearning

---

**Algorithm 1** Machine Unlearning with  Shared Optimizer  /  Dual Optimizers

---

1: **Input:** Model: $f_\theta$; Forget set: $\mathcal{D}_f$; Retain set: $\mathcal{D}_r$; Iterations for outer loop: $T_o$; Iterations for forgetting: $T_f$; Iterations for retaining: $T_r$; Step sizes:  $\eta$ ,  $\eta_f, \eta_r$ .

2:  Optim is the same optimizer as in pretraining with step size $\eta$.

 $\mathrm{Optim}_f$ is $\mathrm{Adam}(\theta, \eta_f)$, $\mathrm{Optim}_r$ is the same optimizer as in pretraining with step size $\eta_r$.

3: **for** $t = 1, ..., T_o$ **do**
4:     **for** $t' = 1, ..., T_f$ **do**
5:         Fetch mini-batch data from the forget set $B_f \sim \mathcal{D}_f$
6:         Calculate the forget loss $\mathcal{L}_f$ on $B_f$ and get the gradient
7:         Use  Optim  /  $\mathrm{Optim}_f$  to update $\theta$
8:     **end for**
9:     **for** $t' = 1, ..., T_r$ **do**
10:        Fetch mini-batch data from the retain set $B_r \sim \mathcal{D}_r$
11:        Calculate the retain loss $\mathcal{L}_r$ on $B_r$ and get the gradient
12:        Use  Optim  /  $\mathrm{Optim}_r$  to update $\theta$
13:     **end for**
14: **end for**
15: **Output:** Model $f_\theta$

---

We incorporate the findings of the two subsections above and propose Dual Optimizers (**DualOptim**) to enhance the efficacy and stability of MU. Specifically, we utilize *two distinct optimizers* during the unlearning process: Adam for forgetting and the default optimizer (e.g., SGD) for retaining. On one hand, we use the same optimizer as in pretraining to maintain the performance on the retain set and use the optimizer with adaptive learning rates to handle the large gradient variation for the forget loss. On the other hand, we decouple the momentum factors and use two distinct optimizers to boost the algorithm stability during unlearning. Notably, DualOptim is a plug-and-play approach and can be integrated in existing MU algorithms that minimize forgetting and retaining objectives alternately. The detailed pseudo-code to compare the pipeline of MU with a shared optimizer and DualOptim is presented in Algorithm 1, with unique segments for shared and dual optimizers highlighted in red and green, respectively. Algorithm 1 presents a general framework, $\mathcal{L}_f$ and $\mathcal{L}_r$ are specified by different concrete MU algorithms.

## 4 Experiments

In this section, we conduct extensive experiments to evaluate our method for different applications, including image classification, image generation, and large language models. The results demonstrate that our method can enhance the efficacy and stability of multiple MU methods, achieving new state-of-the-art performance across diverse scenarios. We conduct ablation studies for further analysis.

## 4.1 Random Subset Unlearning in Image Classification

Table 1: Performance summary of MU methods for image classification. Experiments are conducted on (a) 10% random subset of **CIFAR-10** using **ResNet-18** and (b) 10% random subset of **TinyImageNet** using **Swin-T**. All results are presented as mean and standard deviation across 5 trials with different random forget data. Performance gaps with *RT* are indicated in blue. The average gap (**Gap**) and average standard deviation (**Std**) metrics are calculated by the average of the gaps and standard deviation measured in FA, RA, TA, and MIA, respectively. All the numbers are in percentage.

**(a) CIFAR-10 Random Subset Unlearning (10%)**

| Method | FA | RA | TA | MIA | Gap ↓ | Std ↓ |
|---|---|---|---|---|---|---|
| RT | $94.61_{\pm 0.46}$ (0.00) | $100.00_{\pm 0.00}$ (0.00) | $94.25_{\pm 0.18}$ (0.00) | $76.26_{\pm 0.54}$ (0.00) | 0.00 | 0.30 |
| FT | $99.16_{\pm 0.10}$ (4.55) | $99.84_{\pm 0.06}$ (0.16) | $94.10_{\pm 0.09}$ (0.15) | $88.77_{\pm 0.38}$ (12.51) | 4.34 | 0.16 |
| GA | $98.76_{\pm 0.39}$ (4.15) | $99.10_{\pm 0.90}$ (0.90) | $93.89_{\pm 0.41}$ (0.36) | $92.58_{\pm 0.55}$ (16.32) | 5.43 | 0.44 |
| RL | $97.19_{\pm 0.21}$ (2.58) | $99.67_{\pm 0.08}$ (0.33) | $94.03_{\pm 0.27}$ (0.22) | $68.19_{\pm 0.95}$ (8.43) | 2.80 | 0.38 |
| SCRUB | $92.88_{\pm 0.25}$ (1.73) | $99.62_{\pm 0.10}$ (0.38) | $93.54_{\pm 0.22}$ (0.71) | $82.78_{\pm 0.86}$ (6.52) | 2.33 | **0.36** |
| +DualOptim | $94.90_{\pm 0.42}$ (0.29) | $99.52_{\pm 0.09}$ (0.48) | $93.50_{\pm 0.20}$ (0.75) | $78.26_{\pm 0.79}$ (2.00) | **0.88** | 0.38 |
| SalUn | $96.99_{\pm 0.31}$ (2.38) | $99.40_{\pm 0.28}$ (0.60) | $93.84_{\pm 0.36}$ (0.41) | $65.76_{\pm 1.05}$ (10.50) | 3.47 | 0.50 |
| +DualOptim | $95.47_{\pm 0.22}$ (0.86) | $99.06_{\pm 0.94}$ (0.60) | $92.47_{\pm 0.29}$ (1.78) | $76.14_{\pm 0.70}$ (0.12) | **0.93** | **0.35** |
| SFRon | $94.67_{\pm 3.03}$ (0.06) | $99.83_{\pm 0.13}$ (0.17) | $93.98_{\pm 0.56}$ (0.27) | $77.80_{\pm 5.61}$ (1.54) | 0.51 | 2.33 |
| +DualOptim | $94.69_{\pm 1.13}$ (0.08) | $99.92_{\pm 0.01}$ (0.08) | $94.11_{\pm 0.11}$ (0.14) | $77.77_{\pm 1.39}$ (1.51) | **0.44** | **0.66** |

**(b) TinyImageNet Random Subset Unlearning (10%)**

| Method | FA | RA | TA | MIA | Gap ↓ | Std ↓ |
|---|---|---|---|---|---|---|
| RT | $85.29_{\pm 0.09}$ (0.00) | $99.55_{\pm 0.03}$ (0.00) | $85.49_{\pm 0.15}$ (0.00) | $69.30_{\pm 0.20}$ (0.00) | 0.00 | 0.12 |
| FT | $96.50_{\pm 0.10}$ (11.21) | $98.23_{\pm 0.08}$ (1.32) | $82.67_{\pm 0.21}$ (2.82) | $79.85_{\pm 0.13}$ (10.55) | 6.48 | 0.13 |
| GA | $90.02_{\pm 3.26}$ (4.73) | $90.84_{\pm 3.29}$ (8.71) | $75.64_{\pm 2.67}$ (9.85) | $78.97_{\pm 2.07}$ (9.67) | 8.24 | 2.82 |
| RL | $94.66_{\pm 0.26}$ (9.37) | $98.02_{\pm 0.14}$ (1.53) | $82.73_{\pm 0.27}$ (2.76) | $54.45_{\pm 1.04}$ (15.15) | 7.13 | 0.43 |
| SCRUB | $97.80_{\pm 0.16}$ (12.51) | $98.13_{\pm 0.08}$ (1.42) | $82.64_{\pm 0.19}$ (2.85) | $79.62_{\pm 0.41}$ (10.32) | 6.78 | **0.21** |
| +DualOptim | $97.20_{\pm 0.20}$ (11.91) | $98.30_{\pm 0.10}$ (1.25) | $83.17_{\pm 0.19}$ (2.32) | $79.10_{\pm 0.63}$ (9.80) | **6.32** | 0.28 |
| SalUn | $97.69_{\pm 0.14}$ (12.40) | $98.89_{\pm 0.03}$ (0.66) | $84.02_{\pm 0.32}$ (1.47) | $61.87_{\pm 0.97}$ (7.43) | 5.49 | 0.37 |
| +DualOptim | $91.68_{\pm 0.28}$ (6.39) | $95.13_{\pm 0.18}$ (4.42) | $80.16_{\pm 0.34}$ (5.33) | $72.48_{\pm 0.33}$ (3.18) | **4.83** | **0.28** |
| SFRon | $96.41_{\pm 0.74}$ (11.12) | $98.95_{\pm 0.22}$ (0.60) | $83.40_{\pm 0.51}$ (2.09) | $70.40_{\pm 3.15}$ (1.10) | 3.73 | 1.16 |
| +DualOptim | $92.26_{\pm 1.44}$ (6.97) | $98.27_{\pm 0.12}$ (1.28) | $83.12_{\pm 0.21}$ (2.37) | $69.19_{\pm 2.27}$ (0.11) | **2.68** | **1.01** |

We start with random subset unlearning tasks in image classification, including using ResNet-18 [45] on CIFAR-10 [46] and Swin Transformer-Tiny (Swin-T) [47] on TinyImageNet [48]. Additional results on CIFAR-100 [46] and SVHN [49] are included in Appendix C.3. Consistent with previous work [21, 24, 6, 7], we regard Retrain (RT) as the gold standard of MU. Since our proposed DualOptim is plug-and-play, we apply it to SCRUB [8], SalUn [6], and SFRon [7] to validate its efficacy and stability. Additionally, we include three simple baselines, namely Fine-tune (FT) [50], Gradient Ascent (GA) [21] and Random Label (RL) [22], for reference as well. Note that the hyperparameters of all evaluated methods are tuned to their optimal values by sophisticated search, and we defer the implementation details to Appendix B. Besides the metrics mentioned in Sec. 3.1, which includes FA, RA, TA and MIA, we follow the evaluation criteria in [6, 7] to report the average gap (Gap) and average standard deviation (Std) to indicate the overall performance and stability, respectively. They are the average gap with RT and the average standard deviation among the results in FA, RA, TA, and MIA, respectively.

As illustrated in Table 1, SFRon achieves the smallest average gap with RT, yet it exhibits the highest variability in performance across different random forget subsets. This suggests that the optimal hyperparameters vary significantly depending on the specific forget sets used. Despite being more stable, other methods yield suboptimal results in terms of unlearning efficacy. By integrating DualOptim into these MU algorithms, we observe notable enhancements in terms of both unlearning efficacy and performance stability. For example, in the task of unlearning a 10% random subset from CIFAR-10, DualOptim reduces the average standard deviation of SFRon from 2.33% to 0.66%, while narrowing the average gap from 0.51% to 0.44%, achieving the best performance among all. Additionally, DualOptim significantly narrows down the average gap with RT for both SCRUB and SalUn, with improvements of 1.45% and 2.54%, respectively. Consistent results are observed in other datasets and model architectures as well, highlighting the widespread effectiveness of DualOptim in improving both the performance and stability of unlearning processes.

## 4.2 Class-wise Unlearning in Image Generation

Table 2: Class-wise unlearning performance on **CIFAR-10** with **DDPM** and **ImageNet** with **DiT**. The best unlearning performance for each forgetting class is highlighted in **bold** for FA (in %) and FID. Note that the results of SA, SalUn and SFRon are those reported in [7].

| Method | CIFAR-10 Class-wise Unlearning | | | | | | | | | |
|---|---|---|---|---|---|---|---|---|---|---|
| | Automobile | | Cat | | Dog | | Horse | | Truck | |
| | FA ↓ | FID ↓ | FA ↓ | FID ↓ | FA ↓ | FID ↓ | FA ↓ | FID ↓ | FA ↓ | FID ↓ |
| SA | **0.00** | 23.56 | 14.20 | 21.34 | 8.60 | 21.19 | **0.00** | 21.13 | **0.00** | 29.04 |
| SalUn | 0.20 | 21.23 | 1.40 | 20.29 | **0.00** | 20.18 | 0.60 | 20.70 | 0.80 | 20.45 |
| SFRon | **0.00** | 20.70 | 7.40 | **18.44** | 0.20 | 18.89 | **0.00** | 19.93 | **0.00** | 20.61 |
| **+DO** | 0.20 | **19.72** | 1.00 | 19.36 | **0.00** | **18.58** | **0.00** | **18.91** | **0.00** | **17.26** |

| Method | ImageNet Class-wise Unlearning | | | | | | | | | |
|---|---|---|---|---|---|---|---|---|---|---|
| | Cockatoo | | Golden Retriever | | White Wolf | | Arctic Fox | | Otter | |
| | FA ↓ | FID ↓ | FA ↓ | FID ↓ | FA ↓ | FID ↓ | FA ↓ | FID ↓ | FA ↓ | FID ↓ |
| SA | **0.00** | 348.75 | **0.00** | 298.97 | **0.00** | 45.89 | **0.00** | 393.91 | 29.8 | 321.21 |
| SalUn | 91.21 | 18.47 | 46.09 | 25.28 | **0.00** | 15.16 | 45.90 | 408.07 | 87.5 | 19.69 |
| SFRon | **0.00** | **13.59** | **0.00** | 17.76 | **0.00** | 23.28 | **0.00** | 16.12 | **0.00** | 16.43 |
| **+DO** | **0.00** | 17.46 | **0.00** | **14.63** | **0.00** | **14.72** | **0.00** | **14.91** | **0.00** | **14.55** |

Besides classification, we further evaluate our proposed DualOptim in class-wise unlearning tasks in image generation. Following the settings of [7], we conduct experiments using conditional DDPM [26] with the U-Net [51] on CIFAR-10 and the latent diffusion model [28] with Diffusion Transformer (DiT) [29] on ImageNet [48]. We apply our proposed DualOptim to SFRon [7], which exhibit the state-of-the-art unlearning performance in image generation. We also include SA [35] and SalUn [6] as baselines for comparison. Note that we do not apply DualOptim to SA and SalUn due to their joint updating scheme and unstable performance on ImageNet. Nonetheless, we still present the results of SalUn+DO on CIFAR-10 in Appendix C.4. The implementation details can be found in Appendix B. As for the evaluation metrics, we adopt the accuracy of the unlearned models generated images on forgetting classes (FA) by a pre-trained classifier to indicate the forgetting efficacy and Fréchet Inception Distance (FID) [52] to assess the generation capability on the retained classes.

As shown in Table 2 and 3, our method generally enhances the fidelity of generated images for retained classes while ensuring high forgetting efficacy and low variance. For example, it significantly reduces the FID of SFRon for the white wolf category from 23.28 to 14.63, with FA remaining at 0.00%. Importantly, these results demonstrate that DualOptim is effective across various datasets and model architectures, making it suitable for practical applications. To-

Table 3: Overall class-wise unlearning performance on CIFAR-10 with DDPM and ImageNet with DiT. The **mean value** and **standard deviation** of FA (in %) and FID among the classes evaluated in Table 2 are reported.

| Method | CIFAR-10 | | ImageNet | |
|---|---|---|---|---|
| | FA ↓ | FID ↓ | FA ↓ | FID ↓ |
| SA | $4.56_{\pm 5.86}$ | $23.25_{\pm 3.03}$ | $5.96_{\pm 11.92}$ | $281.75_{\pm 122.11}$ |
| SalUn | $0.60_{\pm 0.49}$ | $20.57_{\pm 0.37}$ | $54.14_{\pm 33.32}$ | $97.33_{\pm 155.40}$ |
| SFRon | $1.52_{\pm 2.94}$ | $19.71_{\pm 0.91}$ | $\mathbf{0.00}_{\pm 0.00}$ | $17.44_{\pm 3.22}$ |
| **+DO** | $\mathbf{0.24}_{\pm 0.39}$ | $\mathbf{18.77}_{\pm 0.85}$ | $\mathbf{0.00}_{\pm 0.00}$ | $\mathbf{15.25}_{\pm 1.11}$ |

gether with the findings in Section 4.1, these results highlight the effectiveness and generalizability of DualOptim in diverse computer vision tasks. Generated images from the unlearned model utilizing DualOptim can be found in Appendix C.7. The visualization again indicates that DualOptim achieves effective unlearning while maintaining the generation capability for the retained classes.

## 4.3 Random Subset Unlearning in Large Language Models

Following the success of MU in vision tasks, there is a rising interest of applying it to remove private or harmful information from large language models (LLMs). We conduct the experiments on Phi-1.5 [53] and LLaMA 2 [37], both are fine-tuned on TOFU dataset [2] with 1.3B and 7B parameters, respectively. We include three **untargeted unlearning** methods (GA+GD, NPO+GD and ME+GD) and two **targeted unlearning** methods (DPO+GD and IDK+AP) [9] as the baselines. We employ

Table 4: Performance comparison of different MU methods on TOFU-finetuned **Phi-1.5** and **LLaMA 2**. The results include Model Capability (MC), Forget Efficacy (FE), and the average metric (Avg.) for forget 1%, 5% data, and 10% data.

| | Phi-1.5 | | | | | | | | |
| Method | forget 1% data | | | forget 5% data | | | forget 10% data | | |
| | MC ↑ | FE ↑ | Avg. ↑ | MC ↑ | FE ↑ | Avg. ↑ | MC↑ | FE ↑ | Avg. ↑ |
|---|---|---|---|---|---|---|---|---|---|
| GA+GD | 0.4934 | 0.4493 | 0.4714 | 0.4360 | 0.5084 | 0.4722 | 0.4471 | 0.5246 | 0.4859 |
| NPO+GD | 0.2569 | 0.5682 | 0.4125 | 0.4940 | 0.4469 | 0.4705 | 0.4808 | 0.4382 | 0.4595 |
| ME+GD | **0.4944** | 0.3938 | 0.4441 | 0.4559 | 0.4480 | 0.4520 | 0.4594 | 0.4564 | 0.4579 |
| **+DO** | 0.4866 | **0.6913** | **0.5889** | **0.4676** | **0.8200** | **0.6438** | **0.5009** | **0.7732** | **0.6370** |
| DPO+GD | 0.2410 | 0.6831 | 0.4621 | 0.4105 | 0.6334 | 0.5219 | 0.3517 | 0.6302 | 0.4910 |
| IDK+AP | **0.4403** | 0.5723 | 0.5063 | **0.4800** | 0.5112 | 0.4956 | **0.4614** | 0.6003 | 0.5308 |
| **+DO** | 0.4221 | **0.7037** | **0.5629** | 0.4633 | **0.6974** | **0.5804** | 0.4422 | **0.7193** | **0.5807** |

| | LLaMA 2 | | | | | | | | |
| Method | forget 1% data | | | forget 5% data | | | forget 10% data | | |
| | MC ↑ | FE ↑ | Avg. ↑ | MC ↑ | FE ↑ | Avg. ↑ | MC ↑ | FE ↑ | Avg. ↑ |
|---|---|---|---|---|---|---|---|---|---|
| GA+GD | 0.6696 | 0.5908 | 0.6302 | 0.0000 | 0.8772 | 0.4386 | 0.5592 | 0.9346 | 0.7469 |
| NPO+GD | 0.6414 | 0.6109 | 0.6262 | 0.5465 | 0.6921 | 0.6193 | 0.5648 | 0.7668 | 0.6658 |
| ME+GD | 0.7271 | 0.9204 | 0.8237 | **0.7472** | 0.9313 | 0.8392 | **0.7357** | 0.9489 | 0.8423 |
| **+DO** | **0.7425** | **0.9612** | **0.8519** | 0.7316 | **0.9602** | **0.8459** | 0.7315 | **0.9625** | **0.8470** |
| DPO+GD | 0.7564 | 0.5335 | 0.6450 | 0.0000 | 0.8243 | 0.4122 | 0.0000 | 0.8041 | 0.4021 |
| IDK+AP | **0.7580** | 0.7625 | 0.7603 | **0.7529** | 0.7479 | 0.7504 | **0.7471** | 0.7433 | 0.7452 |
| **+DO** | 0.7412 | **0.8075** | **0.7743** | 0.7354 | **0.7958** | **0.7656** | 0.7362 | **0.7855** | **0.7609** |

DualOptim (DO) in ME+GD and IDK+AP, which perform the best in [9], to validate the effectiveness of our method. Consistent with [2, 9], we consider three levels of unlearning tasks: to forget 1%, 5%, and 10% of the constructed data. We follow [9] and adopt the improved Model Capability (MC) [3] and Forger Efficacy (FE) as the evaluation metrics.

The results in Table 4 indicate that our method significantly enhances both FE and MC in general. Although a slight reduction in MC is observed for some instances, the more effective forgetting achieved ultimately leads to an increase in the average metric for all cases. Note that the standard deviations are omitted since they are smaller than 5% of the corresponding mean values in all cases. In addition, as shown in Figure 7 of Appendix C.5, the unlearning process utilizing DualOptim demonstrates greater effectiveness and stability compared wtih other baselines. These findings underscore the effectiveness of our method in terms of both performance and stability for LLMs.

It is important to note that the forgetting and retaining objectives are jointly optimized in the baselines listed in Table 4, whereas our method alternates between optimizing these two objectives. To ensure a fair comparison, we also evaluate unlearning performance by alternately optimizing the forgetting and retaining objectives using a shared optimizer. The results in Table 15 of Appendix C.5 demonstrate that DualOptim surpasses both the joint and alternate update schemes.

Table 5: Running time and memory usage of Adam and DualOptim on Llama 2 with full-parameter tuning and LoRA. Note that for full-parameter tuning, we used two H20 GPUs and DeepSpeed zero stage 2 is adopted; for LoRA, we used a single H20 GPU.

(a) **Running Time Usage (min)**

| Tuning method | MU method | Adam | DualOptim | Ratio |
|---|---|---|---|---|
| Full-parameter | ME+GD | 19.40 | 28.50 | 1.47 |
| | IDK+AP | 22.77 | 33.92 | 1.49 |
| LoRA | ME+GD | 17.04 | 17.43 | 1.02 |
| | IDK+AP | 26.76 | 62.32 | 1.21 |

(b) **Memory Usage (GB/GPU)**

| Tuning method | MU method | Adam | DualOptim | Ratio |
|---|---|---|---|---|
| Full-parameter | IDK+AP | 59.83 | 86.40 | 1.44 |
| | ME+GD | 59.84 | 86.52 | 1.45 |
| LoRA | IDK+AP | 30.63 | 30.62 | 1.00 |
| | ME+GD | 31.44 | 30.62 | 0.97 |

Although DualOptim exhibits effectiveness in MU for LLMs, as shown in Table 5, the decoupled momentum introduces approximately $1.5\times$ computational and memory overhead because of large amount of parameters in LLMs. To investigate our method's effectiveness under limited computation

---

[3]Model Capability (MC) is referred to Model Utility (MU) in [9], which conflicts with the abbreviation of Machine Unlearning (MU) in this paper.

resources, we apply DualOptim to the parameter-efficient fine-tuning techniques, such as LoRA [54]. As indicated in Table 16 of Appendix C.5, DualOptim achieves a minimal compromise in unlearning performance by using LoRA, while significantly reducing memory consumption on optimizer states.

## 4.4 Ablation Studies

In this subsection, we compare different combinations of optimizers to further analyze the proposed method. More ablation studies can be found in Appendix C.6.

We compare various combinations of the Adam and SGD optimizers for the forget loss and the retain loss with their standalone counterparts. Besides Adam, we also investigate the combinations of SGD and other optimizers, e.g., Lion [55] and Muon [56]. The results in Table 6 indicate that the configuration of Adam (F) + SGD (R) offers the optimal balance between performance and stability. This finding underscores the importance of decoupled momentum and adaptive learning rate for the forgetting objective. Moreover, the results presented in Table 18 and Table 19 in Appendix C.6 indicate that the best optimizer for retaining depends on the choice of optimizer during pretraining and the specific MU algorithms employed. It should be pointed out that a shared SGD demonstrates a competitive average performance but suffers from high standard deviation indicating instability. Conversely, Lion exhibits low variability but does not achieve optimal performance. These insights further emphasize the efficacy of our method in enhancing performance and stability in MU tasks.

Table 6: Ablation study on different optimizers when we use DualOptim in **SFRon**. Results are based on $10\%$ random subset unlearning task on CIFAR-10 using ResNet-18 pre-trained by SGD. (F) and (R) denotes that the optimizer is used to minimize the forget and retain losses, respectively.

| Optimizer | FA | RA | TA | MIA | Gap ↓ | Std ↓ |
|---|---|---|---|---|---|---|
| Single SGD | $94.67_{\pm3.03}$ (0.06) | $99.83_{\pm0.13}$ (0.17) | $93.98_{\pm0.56}$ (0.27) | $77.80_{\pm5.61}$ (1.54) | 0.51 | 2.33 |
| Single Adam | $94.54_{\pm2.41}$ (0.07) | $99.96_{\pm0.02}$ (0.04) | $94.15_{\pm0.30}$ (0.10) | $81.46_{\pm2.42}$ (5.20) | 1.35 | 1.29 |
| SGD (F) + SGD (R) | $94.07_{\pm3.48}$ (0.54) | $99.90_{\pm0.06}$ (0.10) | $93.93_{\pm0.63}$ (0.32) | $78.48_{\pm4.03}$ (2.22) | 0.80 | 2.05 |
| SGD (F) + Adam (R) | $94.58_{\pm3.49}$ (0.03) | $99.38_{\pm0.78}$ (0.62) | $92.84_{\pm1.62}$ (0.14) | $81.13_{\pm4.58}$ (4.87) | 1.73 | 2.62 |
| Adam (F) + Adam (R) | $94.29_{\pm1.23}$ (0.32) | $99.94_{\pm0.01}$ (0.06) | $94.02_{\pm0.11}$ (0.23) | $77.86_{\pm1.39}$ (1.60) | 0.55 | 0.63 |
| Adam (F) + SGD (R) | $94.69_{\pm1.13}$ (0.02) | $99.92_{\pm0.01}$ (0.08) | $94.11_{\pm0.11}$ (0.14) | $77.77_{\pm1.39}$ (1.51) | **0.44** | 0.66 |
| SGD (F) + Lion (R) | $97.88_{\pm1.49}$ (3.27) | $99.81_{\pm0.28}$ (0.19) | $93.86_{\pm0.94}$ (0.39) | $87.62_{\pm2.73}$ (11.36) | 3.80 | 1.36 |
| Lion (F) + Lion (R) | $94.54_{\pm1.59}$ (0.07) | $99.93_{\pm0.02}$ (0.07) | $93.91_{\pm0.29}$ (0.34) | $83.08_{\pm1.35}$ (6.82) | 1.82 | 0.81 |
| Lion (F) + SGD (R) | $94.66_{\pm1.48}$ (0.05) | $99.91_{\pm0.03}$ (0.09) | $93.95_{\pm0.27}$ (0.30) | $79.55_{\pm1.41}$ (3.29) | 0.93 | 0.80 |
| SGD (F) + Muon (R) | $95.74_{\pm0.40}$ (1.13) | $99.61_{\pm0.03}$ (0.39) | $94.04_{\pm0.09}$ (0.21) | $84.03_{\pm0.77}$ (7.77) | 2.37 | 0.32 |
| Muon (F) + Muon (R) | $95.12_{\pm0.54}$ (0.51) | $99.64_{\pm0.01}$ (0.36) | $93.84_{\pm0.10}$ (0.41) | $82.77_{\pm0.60}$ (6.51) | 1.94 | **0.31** |
| Muon (F) + SGD (R) | $94.57_{\pm1.12}$ (0.04) | $99.93_{\pm0.01}$ (0.07) | $94.13_{\pm0.12}$ (0.12) | $72.46_{\pm1.01}$ (3.80) | 1.01 | 0.57 |

## 5 Conclusion

This study improves efficacy and stability in approximate machine unlearning (MU) methods with Dual Optimizers (DualOptim). By integrating adaptive learning rates and decoupled momentum, DualOptim enhances unlearning performance across diverse applications in computer vision and natural language processing. Its plug-and-play design ensures easy integration into existing MU frameworks. DualOptim marks a significant advancement in the MU field, offering an effective solution to meet the demand for trustworthy machine learning systems.

## Broader Impacts and Limitations

Our method contributes to the trustworthiness of deep learning and can be broadly applicable across diverse tasks. Although DualOptim can be integrated with parameter-efficient fine-tuning methods, it still introduces double memory consumption for optimizer states. We leave the development of a more efficient approach as future work.

## Acknowledgments and Disclosure of Funding

This work is supported by National Natural Science Foundation of China (NSFC Project No. 62306250) and City University of Hong Kong (CityU Project No. 9220132).

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

# A Proofs

## A.1 Proof of Lemma 3.4

*Proof.* Let $\theta_1, \theta_2$ be independent copies of $\theta$ with probability density $p(\theta)$. Using the variance identity $\text{Var}(X) = \frac{1}{2}\mathbb{E}[\|X_1 - X_2\|^2]$ for independent copies $X_1, X_2$, we have:

$$\text{Var}(\nabla_\theta \mathcal{L}(\theta)) = \frac{1}{2} \iint p(\theta_1)p(\theta_2) \|\nabla\mathcal{L}(\theta_1) - \nabla\mathcal{L}(\theta_2)\|^2 \, d\theta_1 d\theta_2. \tag{7}$$

By $L$-smoothness, $\|\nabla\mathcal{L}(\theta_1) - \nabla\mathcal{L}(\theta_2)\| \leq L\|\theta_1 - \theta_2\|$, we have:

$$\text{Var}(\nabla_\theta \mathcal{L}(\theta)) \leq \frac{L^2}{2} \iint p(\theta_1)p(\theta_2)\|\theta_1 - \theta_2\|^2 d\theta_1 d\theta_2. \tag{8}$$

Since $\text{Var}(\theta) \leq \sigma_\theta^2$, we have $\frac{1}{2} \iint p(\theta_1)p(\theta_2)\|\theta_1 - \theta_2\|^2 d\theta_1 d\theta_2 \leq \sigma_\theta^2$. Therefore, we can conclude $\text{Var}(\nabla_\theta \mathcal{L}(\theta)) \leq L^2 \sigma_\theta^2$

**Discussion.** Lemma 3.4 is an important tool for us to analyze the variance of the full-batch gradient, because the full-batch gradient is calculated based on the model parameters obtained in the last iteration, which is also a random variable. Specifically, we can derive the following inequality from the update rule in (5) for further analysis. We add no superscripts, indicating it is applicable for both update schemes.

$$\begin{aligned} \text{Var}(\widehat{\boldsymbol{g}}_{f,i}) &\leq \sigma^2 + \text{Var}(\nabla_\theta \mathcal{L}_f(\theta_{r,i-1})) \leq \sigma^2 + L^2 \text{Var}(\theta_{r,i-1}) \\ \text{Var}(\widehat{\boldsymbol{g}}_{r,i}) &\leq \sigma^2 + \text{Var}(\nabla_\theta \mathcal{L}_r(\theta_{f,i})) \leq \sigma^2 + L^2 \text{Var}(\theta_{f,i}) \end{aligned} \tag{9}$$

$\square$

## A.2 Proof of Theorem 3.5

*Proof.* Without the loss of generality, we can assume the pretrained parameter $\theta_o = 0$ and the learning rate $\eta = 1$, because shift and multiplication do not change the inequalities in the conclusion to prove.

Unfold the update rules in (5) and we obtain

$$\text{(Shared Momentum)} \begin{cases} \boldsymbol{m}_{f,t}^S = \alpha\boldsymbol{m}_{r,t-1}^S + \widehat{\boldsymbol{g}}_{f,t}^S = \sum_{i=0}^t \alpha^{2(t-i)}\widehat{\boldsymbol{g}}_{f,i}^S + \sum_{i=0}^{t-1} \alpha^{2(t-i)-1}\widehat{\boldsymbol{g}}_{r,i}^S \\ \boldsymbol{m}_{r,t}^S = \alpha\boldsymbol{m}_{f,t}^S + \widehat{\boldsymbol{g}}_{r,t}^S = \sum_{i=0}^t \alpha^{2(t-i)+1}\widehat{\boldsymbol{g}}_{f,i}^S + \sum_{i=0}^t \alpha^{2(t-i)}\widehat{\boldsymbol{g}}_{r,i}^S \end{cases}$$

$$\text{(Decoupled Momentum)} \begin{cases} \boldsymbol{m}_{f,t}^D = \alpha\boldsymbol{m}_{f,t-1}^D + \widehat{\boldsymbol{g}}_{f,t}^D = \sum_{i=0}^t \alpha^{t-i}\widehat{\boldsymbol{g}}_{f,i}^D \\ \boldsymbol{m}_{r,t}^D = \alpha\boldsymbol{m}_{r,t-1}^D + \widehat{\boldsymbol{g}}_{r,t}^D = \sum_{i=0}^t \alpha^{t-i}\widehat{\boldsymbol{g}}_{r,i}^D \end{cases} \tag{10}$$

For notation simplicity, we let $A_k = \sum_{i=0}^k \alpha^i$. It is clear that $\forall k_1 \leq k_2, A_{k_1} \leq A_{k_2}$.

Based on the update scheme (5) and $\eta = 1$, we have:

$$\text{(Shared Momentum)} \begin{cases} -\theta_{f,t}^S = \sum_{i=0}^t \boldsymbol{m}_{f,i}^S + \sum_{i=0}^{t-1} \boldsymbol{m}_{r,i}^S \\ -\theta_{r,t}^S = \sum_{i=0}^t \boldsymbol{m}_{f,i}^S + \sum_{i=0}^t \boldsymbol{m}_{r,i}^S \end{cases}$$

$$\text{(Decoupled Momentum)} \begin{cases} -\theta_{f,t}^D = \sum_{i=0}^t \boldsymbol{m}_{f,i}^D + \sum_{i=0}^{t-1} \boldsymbol{m}_{r,i}^D \\ -\theta_{r,t}^D = \sum_{i=0}^t \boldsymbol{m}_{f,i}^D + \sum_{i=0}^t \boldsymbol{m}_{r,i}^D \end{cases} \tag{11}$$

Combining (10) and (11), we have the following equations. For notation simplicity, we let $A_k = \sum_{i=0}^k \alpha^i$. It is clear that $\forall k_1 \leq k_2, A_{k_1} \leq A_{k_2}$.

$$\text{(Shared Momentum)} \begin{cases} -\theta_{f,t}^S = \sum_{i=0}^t A_{2(t-i)}\widehat{\boldsymbol{g}}_{f,i}^S + \sum_{i=0}^{t-1} A_{2(t-i)-1}\widehat{\boldsymbol{g}}_{r,i}^S \\ -\theta_{r,t}^S = \sum_{i=0}^t A_{2(t-i)+1}\widehat{\boldsymbol{g}}_{f,i}^S + \sum_{i=0}^t A_{2(t-i)}\widehat{\boldsymbol{g}}_{r,i}^S \end{cases}$$

$$\text{(Decoupled Momentum)} \begin{cases} -\theta_{f,t}^D = \sum_{i=0}^t A_{t-i}\widehat{\boldsymbol{g}}_{f,i}^D + \sum_{i=0}^{t-1} A_{t-i-1}\widehat{\boldsymbol{g}}_{r,i}^D \\ -\theta_{r,t}^D = \sum_{i=0}^t A_{t-i}\widehat{\boldsymbol{g}}_{f,i}^D + \sum_{i=0}^t A_{t-i}\widehat{\boldsymbol{g}}_{r,i}^D \end{cases} \tag{12}$$

**We prove the theorem by mathematical induction.**

**We start with the case of** $t = 0$. Based on (12), we have $\theta_{f,0}^S = -\widehat{g}_{f,0}^S$, $\theta_{f,0}^D = -\widehat{g}_{f,0}^D$. Both are stochastic gradients calculated on the initial parameter $\theta_o$, so based on Assumption 3.1, we have:

$$\mathrm{Var}(\theta_{f,0}^S) \le \sigma^2 \overset{\text{def}}{=} \overline{\mathrm{Var}}(\theta_{f,0}^S), \quad \mathrm{Var}(\theta_{f,0}^D) \le \sigma^2 \overset{\text{def}}{=} \overline{\mathrm{Var}}(\theta_{f,0}^D). \tag{13}$$

In addition, we have $\theta_{r,0}^D = -\widehat{g}_{f,0}^D - \widehat{g}_{r,0}^D$ and $\theta_{r,0}^S = -(1+\alpha)\widehat{g}_{f,0}^S - \widehat{g}_{r,0}^S$ based on (12). Based on Lemma 3.4 and inequality (9), we have:

$$\begin{aligned}
\mathrm{Var}(\widehat{g}_{r,0}^S) &\le \sigma^2 + L^2\mathrm{Var}(\theta_{f,0}^S) \le (L^2+1)\sigma^2, \\
\mathrm{Var}(\widehat{g}_{r,0}^D) &\le \sigma^2 + L^2\mathrm{Var}(\theta_{f,0}^D) \le (L^2+1)\sigma^2.
\end{aligned} \tag{14}$$

We assume negative correlation between gradients from the forget loss and the retain loss in Assumption 3.2. Therefore, we can derive the variance for $\theta_{r,0}^S$ and $\theta_{r,0}^D$ as follows:

$$\begin{aligned}
\mathrm{Var}(\theta_{r,0}^S) &\le (1+\alpha)^2\mathrm{Var}(\widehat{g}_{f,0}^S) + \mathrm{Var}(\widehat{g}_{r,0}^S) \le (1+\alpha)^2\sigma^2 + (L^2+1)\sigma^2 \overset{\text{def}}{=} \overline{\mathrm{Var}}(\theta_{r,0}^S), \\
\mathrm{Var}(\theta_{r,0}^D) &\le \mathrm{Var}(\widehat{g}_{f,0}^D) + \mathrm{Var}(\widehat{g}_{r,0}^D) \le \sigma^2 + (1+L^2)\sigma^2 \overset{\text{def}}{=} \overline{\mathrm{Var}}(\theta_{r,0}^D).
\end{aligned} \tag{15}$$

Thus, we have the following conclusion:

$$\overline{\mathrm{Var}}(\theta_{f,0}^D) \le \overline{\mathrm{Var}}(\theta_{f,0}^S), \quad \overline{\mathrm{Var}}(\theta_{r,0}^D) \le \overline{\mathrm{Var}}(\theta_{r,0}^S). \tag{16}$$

**That is to say, the conclusion of the theorem holds for** $t = 0$. **Now we assume that the conclusion of the theorem holds for** $0, 1, ..., t - 1$, **and then consider the case of** $t$.

Based on Assumption 3.2, we have the following inequality for both update schemes:

$$\begin{aligned}
\forall w_0, ..., w_t, \mathrm{Var}\left(\sum_{i=0}^t w_i\widehat{g}_{f,i}\right) &= \sum_{i=0}^t w_i^2\mathrm{Var}(\widehat{g}_{f,i}) + 2\sum_{i\neq j}\rho(\widehat{g}_{f,i}, \widehat{g}_{f,j})w_iw_j\sqrt{\mathrm{Var}(\widehat{g}_{f,i})\mathrm{Var}(\widehat{g}_{f,j})} \\
&\le \sum_{i=0}^t w_i^2\mathrm{Var}(\widehat{g}_{f,i}) + \sum_{i\neq j}\tau\left(w_i^2\mathrm{Var}(\widehat{g}_{f,i}) + w_j^2\mathrm{Var}(\widehat{g}_{f,j})\right) = (1+t\tau)\sum_{i=0}^t w_i^2\mathrm{Var}(\widehat{g}_{f,i})
\end{aligned} \tag{17}$$

Similarly, we have $\forall w_0, ..., w_t, \mathrm{Var}\left(\sum_{i=0}^t w_i\widehat{g}_{r,i}\right) \le (1+t\tau)\sum_{i=0}^t w_i^2\mathrm{Var}(\widehat{g}_{r,i})$.

Now we combine (12), (14), (17) and can derive the following inequalities:

$$\begin{aligned}
\mathrm{Var}(\theta_{f,t}^S) &\le (1+t\tau)\sum_{i=0}^t A_{2(t-i)}^2\left(\sigma^2 + L^2\mathrm{Var}(\theta_{r,i-1}^S)\right) + (1+(t-1)\tau)\sum_{i=0}^{t-1} A_{2(t-i)-1}^2\left(\sigma^2 + L^2\mathrm{Var}(\theta_{f,i}^S)\right) \\
\mathrm{Var}(\theta_{f,t}^D) &\le (1+t\tau)\sum_{i=0}^t A_{t-i}^2\left(\sigma^2 + L^2\mathrm{Var}(\theta_{r,i-1}^D)\right) + (1+(t-1)\tau)\sum_{i=0}^{t-1} A_{t-i-1}^2\left(\sigma^2 + L^2\mathrm{Var}(\theta_{f,i}^D)\right) \\
\mathrm{Var}(\theta_{r,t}^S) &\le (1+t\tau)\sum_{i=0}^t A_{2(t-i)+1}^2\left(\sigma^2 + L^2\mathrm{Var}(\theta_{r,i-1}^S)\right) + (1+t\tau)\sum_{i=0}^t A_{2(t-i)}^2\left(\sigma^2 + L^2\mathrm{Var}(\theta_{f,i}^S)\right) \\
\mathrm{Var}(\theta_{r,t}^D) &\le (1+t\tau)\sum_{i=0}^t A_{t-i}^2\left(\sigma^2 + L^2\mathrm{Var}(\theta_{r,i-1}^D)\right) + (1+t\tau)\sum_{i=0}^t A_{t-i}^2\left(\sigma^2 + L^2\mathrm{Var}(\theta_{f,i}^D)\right)
\end{aligned} \tag{18}$$

By definition, $\overline{\mathrm{Var}}(\theta_{f,t}^D)$, $\overline{\mathrm{Var}}(\theta_{f,t}^S)$, $\overline{\mathrm{Var}}(\theta_{r,t}^D)$, and $\overline{\mathrm{Var}}(\theta_{r,t}^S)$ are the maximum possible value of $\mathrm{Var}(\theta_{f,t}^D)$, $\mathrm{Var}(\theta_{f,t}^S)$, $\mathrm{Var}(\theta_{r,t}^D)$, and $\mathrm{Var}(\theta_{r,t}^S)$, respectively. Considering the inequalities in (18) are all achievable, we have the following:

$$\overline{\mathrm{Var}}(\theta_{f,t}^S) = (1+t\tau)\sum_{i=0}^{t} A_{2(t-i)}^2 \left(\sigma^2 + L^2\overline{\mathrm{Var}}(\theta_{r,i-1}^S)\right) + (1+(t-1)\tau)\sum_{i=0}^{t-1} A_{2(t-i)-1}^2 \left(\sigma^2 + L^2\overline{\mathrm{Var}}(\theta_{f,i}^S)\right)$$

$$\overline{\mathrm{Var}}(\theta_{f,t}^D) = (1+t\tau)\sum_{i=0}^{t} A_{t-i}^2 \left(\sigma^2 + L^2\overline{\mathrm{Var}}(\theta_{r,i-1}^D)\right) + (1+(t-1)\tau)\sum_{i=0}^{t-1} A_{t-i-1}^2 \left(\sigma^2 + L^2\overline{\mathrm{Var}}(\theta_{f,i}^D)\right)$$

$$\overline{\mathrm{Var}}(\theta_{r,t}^S) = (1+t\tau)\sum_{i=0}^{t} A_{2(t-i)+1}^2 \left(\sigma^2 + L^2\overline{\mathrm{Var}}(\theta_{r,i-1}^S)\right) + (1+t\tau)\sum_{i=0}^{t} A_{2(t-i)}^2 \left(\sigma^2 + L^2\overline{\mathrm{Var}}(\theta_{f,i}^S)\right)$$

$$\overline{\mathrm{Var}}(\theta_{r,t}^D) = (1+t\tau)\sum_{i=0}^{t} A_{t-i}^2 \left(\sigma^2 + L^2\overline{\mathrm{Var}}(\theta_{r,i-1}^D)\right) + (1+t\tau)\sum_{i=0}^{t} A_{t-i}^2 \left(\sigma^2 + L^2\overline{\mathrm{Var}}(\theta_{f,i}^D)\right)$$

$$(19)$$

**By induction, we have for $t' \in \{0,1,...,t-1\}$, $\overline{\mathrm{Var}}(\theta_{f,t'}^D) \le \overline{\mathrm{Var}}(\theta_{f,t'}^S)$, $\overline{\mathrm{Var}}(\theta_{r,t'}^D) \le \overline{\mathrm{Var}}(\theta_{r,t'}^S)$. In addition, $\forall k_1 \le k_2$, $A_{k_1} \le A_{k_2}$. Therefore, after comparing the factors and terms on the right hand of (19), it is obvious to obtain the following conclusion.**

$$\overline{\mathrm{Var}}(\theta_{f,t}^D) \le \overline{\mathrm{Var}}(\theta_{f,t}^S), \quad \overline{\mathrm{Var}}(\theta_{r,t}^D) \le \overline{\mathrm{Var}}(\theta_{r,t}^S). \tag{20}$$

**By induction, the conclusion of this theorem holds for any $t \ge 0$.** $\qquad\square$

### A.3 Variance Bound of Performance Metric Function

We let $M(\theta)$ to represent the performance of model with parameter $\theta$ where $M$ can be FA/RA/TA/MIA. Before we derive the variance bound of performance metric function $M(\theta)$, we make the following assumption:

**Assumption A.1.** The performance metric function $M$ is Lipschitz continuous:

$$\forall \theta_1, \theta_2, \|M(\theta_1) - M(\theta_2)\| \le L_M \|\theta_1 - \theta_2\|, \tag{21}$$

where $L_M$ is a non-negative constant.

Following a similar logic to Lemma 3.4, we have the following corollary:

**Corollary A.2.** *If Assumption A.1 holds, and the parameter variance satisfies $\mathrm{Var}(\theta) \le \overline{\mathrm{Var}}(\theta)$, then we have $\mathrm{Var}(M(\theta)) \le L_M^2 \overline{\mathrm{Var}}(\theta)$.*

*Proof.* Similar to the proof of Lemma 3.4, we have:

$$\mathrm{Var}(M(\theta)) = \frac{1}{2}\iint p(\theta_1)p(\theta_2)\|M(\theta_1) - M(\theta_2)\|^2 \, d\theta_1 d\theta_2, \tag{22}$$

where $\theta_1, \theta_2$ are independent copies of $\theta$ with probability density $p(\theta)$.

By $\|M(\theta_1) - M(\theta_2)\| \le L_M\|\theta_1 - \theta_2\|$, we have:

$$\mathrm{Var}(M(\theta)) \le \frac{L_M^2}{2}\iint p(\theta_1)p(\theta_2)\|\theta_1 - \theta_2\|^2 d\theta_1 d\theta_2. \tag{23}$$

Since $\mathrm{Var}(\theta) \le \overline{\mathrm{Var}}(\theta)$, we have $\frac{1}{2}\iint p(\theta_1)p(\theta_2)\|\theta_1 - \theta_2\|^2 d\theta_1 d\theta_2 \le \overline{\mathrm{Var}}(\theta)$. Therefore, we can conclude $\mathrm{Var}(M(\theta)) \le L_M^2 \overline{\mathrm{Var}}(\theta)$. $\qquad\square$

**Discussion.** The final unlearned parameters are usually the parameter updated by retaining loss at the last iteration $T$, i.e., $\theta_{r,T}^D$ using decoupled momentum and $\theta_{r,T}^S$ using shared momentum. According to Theorem 3.5, we have $\overline{\mathrm{Var}}(\theta_{r,T}^D) \le \overline{\mathrm{Var}}(\theta_{r,T}^S)$. Let the upper bound of the performance variance $\overline{\mathrm{Var}}(M(\theta)) = L_M^2 \overline{\mathrm{Var}}(\theta)$, we can derive $\overline{\mathrm{Var}}(M(\theta_{r,T}^D)) \le \overline{\mathrm{Var}}(M(\theta_{r,T}^S))$. Thus, **decoupled momentum can also induce less variability on performance metrics**.

# B Implementation Details

## B.1 Implementation Details for Image Classification

For **CIFAR-10**, **CIFAR-100**, and **SVHN** using **ResNet-18**, all baselines use the SGD optimizer with momentum of 0.9, weight decay of $5 \times 10^{-4}$, and batch size of 128 if not specified. For **TinyImageNet**, **Swin-T**, the models are initialized from torchvision weight pre-trained on ImageNet. All baselines use the **Adam** optimizer with a weight decay of $5 \times 10^{-4}$ and batch size of 128 if not specified. Summaries of the hyperparameters for each method on each dataset are shown in Table 7 - 10. Note that the unspecified hyperparameters are the same as the default ones reported in their original papers.

Table 7: Summary of hyperparameters for each method on unlearning 10% random subset of CIFAR-10. $\eta$ is short for learning rate.

| Method | Hyperparameters |
|---|---|
| Pretrain | epoch = 200, cosine scheduler, $\eta = 0.1$ |
| RT | epoch = 200, cosine scheduler, $\eta = 0.1$ |
| FT | epoch = 10, constant scheduler, $\eta = 0.01$ |
| GA | epoch = 10, constant scheduler, $\eta = 5 \times 10^{-4}$ |
| RL | epoch = 10, constant scheduler, $\eta_f = 0.01, \eta_r = 0.01$ |
| SCRUB | epoch = 10, constant scheduler, Adam, $\eta_f = 1 \times 10^{-4}, \eta_r = 1 \times 10^{-4}$ |
| SalUn | epoch = 10, constant scheduler, $\eta_f = 0.018, \eta_r = 0.01$ |
| SFRon | $T_{out} = 1500, T_{in} = 5$, cosine scheduler, $\eta_f = 0.01, \eta_r = 0.01, \alpha = 31$ |
| SCRUB+DO | epoch = 10, constant scheduler, Adam (F) + Adam (R), $\eta_f = 1.5 \times 10^{-4}, \eta_r = 1 \times 10^{-4}$ |
| SalUn+DO | epoch = 10, constant scheduler, Adam (F) + SGD (R), $\eta_f = 1.3 \times 10^{-4}, \eta_r = 0.01$ |
| SFRon+DO | $T_{out} = 1500, T_{in} = 5$, cosine scheduler, Adam (F) + SGD (R), $\eta_f = 1 \times 10^{-4}, \eta_r = 0.01, \alpha = 1$ |

Table 8: Summary of hyperparameters for each method on unlearning 50% random subset of CIFAR-10. $\eta$ is short for learning rate.

| Method | Hyperparameters |
|---|---|
| Pretrain | epoch = 200, cosine scheduler, $\eta = 0.1$ |
| RT | epoch = 200, cosine scheduler, $\eta = 0.1$ |
| FT | epoch = 10, constant scheduler, $\eta = 0.01$ |
| GA | epoch = 10, constant scheduler, $\eta = 8 \times 10^{-5}$ |
| RL | epoch = 10, constant scheduler, $\eta_f = 0.01, \eta_r = 0.01$ |
| SCRUB | epoch = 10, constant scheduler, Adam, $\eta_f = 1.6 \times 10^{-5}, \eta_r = 1 \times 10^{-4}$ |
| SalUn | epoch = 10, constant scheduler, $\eta_f = 2.5 \times 10^{-4}, \eta_r = 0.01$ |
| SFRon | $T_{out} = 1500, T_{in} = 5$, cosine scheduler, $\eta_f = 0.01, \eta_r = 0.01, \alpha = 82$ |
| SCRUB+DO | epoch = 10, constant scheduler, Adam (F) + Adam (R), $\eta_f = 2.8 \times 10^{-5}, \eta_r = 1 \times 10^{-4}$ |
| SalUn+DO | epoch = 10, constant scheduler, Adam (F) + SGD (R), $\eta_f = 5.5 \times 10^{-5}, \eta_r = 0.01$ |
| SFRon+DO | $T_{out} = 1500, T_{in} = 5$, cosine scheduler, Adam (F) + SGD (R), $\eta_f = 4.1 \times 10^{-4}, \eta_r = 0.01, \alpha = 1$ |

Table 9: Summary of hyperparameters for each method on unlearning 10% random subset of CIFAR-100. $\eta$ is short for learning rate.

| Method | Hyperparameters |
|---|---|
| Pretrain | epoch = 200, cosine scheduler, $\eta = 0.1$ |
| RT | epoch = 200, cosine scheduler, $\eta = 0.1$ |
| FT | epoch = 10, constant scheduler, $\eta = 4.5 \times 10^{-2}$ |
| GA | epoch = 10, constant scheduler, $\eta = 5.2 \times 10^{-4}$ |
| RL | epoch = 10, constant scheduler, $\eta_f = 8 \times 10^{-4}, \eta_r = 0.01$ |
| SCRUB | epoch = 10, constant scheduler, Adam, $\eta_f = 6 \times 10^{-5}, \eta_r = 1 \times 10^{-4}$ |
| SalUn | epoch = 10, constant scheduler, $\eta_f = 1.3 \times 10^{-3}, \eta_r = 0.01$ |
| SFRon | $T_{out} = 1500, T_{in} = 5$, cosine scheduler, $\eta_f = 0.01, \eta_r = 0.01, \alpha = 44$ |
| SCRUB+DO | epoch = 10, constant scheduler, Adam (F) + Adam (R), $\eta_f = 1 \times 10^{-4}, \eta_r = 1 \times 10^{-4}$ |
| SalUn+DO | epoch = 10, constant scheduler, Adam (F) + SGD (R), $\eta_f = 1.9 \times 10^{-4}, \eta_r = 0.01$ |
| SFRon+DO | $T_{out} = 1500, T_{in} = 5$, cosine scheduler, Adam (F) + SGD (R), $\eta_f = 1 \times 10^{-4}, \eta_r = 0.01, \alpha = 1$ |

Table 10: Summary of hyperparameters for each method on unlearning 10% random subset of SVHN. $\eta$ is short for learning rate.

| Method | Hyperparameters |
|---|---|
| Pretrain | epoch $= 200$, cosine scheduler, $\eta = 0.1$ |
| RT | epoch $= 200$, cosine scheduler, $\eta = 0.1$ |
| FT | epoch $= 10$, constant scheduler, $\eta = 2.2 \times 10^{-2}$ |
| GA | epoch $= 10$, constant scheduler, $\eta = 5.2 \times 10^{-4}$ |
| RL | epoch $= 10$, constant scheduler, $\eta_f = 8 \times 10^{-3}$, $\eta_r = 0.01$ |
| SCRUB | epoch $= 10$, constant scheduler, Adam, $\eta_f = 8 \times 10^{-6}$, $\eta_r = 1 \times 10^{-4}$ |
| SalUn | epoch $= 10$, constant scheduler, $\eta_f = 1.25 \times 10^{-2}$, $\eta_r = 0.01$ |
| SFRon | $T_{out} = 1500$, $T_{in} = 5$, cosine scheduler, $\eta_f = 0.01$, $\eta_r = 0.01$, $\alpha = 12.5$ |
| SCRUB+DO | epoch $= 10$, constant scheduler, Adam (F) + Adam (R), $\eta_f = 2.2 \times 10^{-5}$, $\eta_r = 1 \times 10^{-4}$ |
| SalUn+DO | epoch $= 10$, constant scheduler, Adam (F) + SGD (R), $\eta_f = 2.5 \times 10^{-6}$, $\eta_r = 0.01$ |
| SFRon+DO | $T_{out} = 1500$, $T_{in} = 5$, cosine scheduler, Adam (F) + SGD (R), $\eta_f = 1.6 \times 10^{-4}$, $\eta_r = 0.01$, $\alpha = 1$ |

Table 11: Summary of hyperparameters for each method on unlearning 10% random subset of TinyImageNet. $\eta$ is short for learning rate.

| Method | Hyperparameters |
|---|---|
| Pretrain | epoch $= 10$, cosine scheduler, $\eta = 1 \times 10^{-4}$ |
| RT | epoch $= 10$, cosine scheduler, $\eta = 1 \times 10^{-4}$ |
| FT | epoch $= 1$, constant scheduler, $\eta = 1 \times 10^{-4}$ |
| GA | epoch $= 1$, constant scheduler, $\eta = 5 \times 10^{-6}$ |
| RL | epoch $= 1$, constant scheduler, $\eta_f = 5 \times 10^{-5}$, $\eta_r = 1 \times 10^{-4}$ |
| SCRUB | epoch $= 1$, constant scheduler, Adam, $\eta_f = 2 \times 10^{-5}$, $\eta_r = 2 \times 10^{-5}$ |
| SalUn | epoch $= 1$, constant scheduler, $\eta_f = 9 \times 10^{-5}$, $\eta_r = 9 \times 10^{-5}$ |
| SFRon | $T_{out} = 500$, $T_{in} = 1$, cosine scheduler, $\eta_f = 3 \times 10^{-5}$, $\eta_r = 3 \times 10^{-5}$, $\alpha = 500$ |
| SCRUB+DO | epoch $= 1$, constant scheduler, Adam (F) + Adam (R), $\eta_f = 2 \times 10^{-5}$, $\eta_r = 2 \times 10^{-5}$ |
| SalUn+DO | epoch $= 1$, constant scheduler, Adam (F) + Adam (R), $\eta_f = 9 \times 10^{-5}$, $\eta_r = 9 \times 10^{-5}$ |
| SFRon+DO | $T_{out} = 500$, $T_{in} = 2$, cosine scheduler, Adam (F) + Adam (R), $\eta_f = 1.9 \times 10^{-4}$, $\eta_r = 1.9 \times 10^{-4}$, $\alpha = 1$ |

## B.2 Implementation Details for Image Generation

For **CIFAR-10**, we use **DDPM** based on U-Net architecture with 1000 timesteps for linear $\beta$ schedule. All methods use Adam optimizer and batch size of 128. The hyperparameters of SalUn+DO are: $T_{out} = 150$, $T_{in} = 1$, $\eta_f = 4 \times 10^{-5}$, $\eta_r = 1 \times 10^{-4}$, threshold = top-50%. The hyperparameters of SFRon+DO are: $T_{out} = 150$, $T_{in} = 1$, $\eta_f = 1 \times 10^{-4}$, $\eta_r = 1 \times 10^{-4}$, $\alpha = 1$, $\lambda = 0.5$, $\gamma = 3$. The hyperparameters of other methods are the same as those reported in [7].

For **ImageNet**, we use pre-trained **DiT-XL**/22 with $256 \times 256$ resolution. All methods use AdamW optimizer and batch size of 1. The hyperparameters of SFRon+DO are: $T_{out} = 500$, $T_{in} = 1$, $\eta_f = 5 \times 10^{-5}$, $\eta_r = 5 \times 10^{-5}$, $\alpha = 1$, $\gamma = 3$. Since the batch size is 1, we ignore $\lambda$ in adaptive coefficients. The hyperparameters of other methods are the same as those reported in [7].

## B.3 Implementation Details for Natural Language Processing

For **Phi-1.5** and **LLaMA 2**, we utilize pre-trained models on the **TOFU** dataset [2] and conduct evaluations accordingly. For the baseline methods, including **GA+GD** and **DPO+GD**, as well as the methods **ME+GD** and **IDK+AP**, we adopt the default hyperparameters reported in [9]. For **IDK+AP**, the forget coefficient and regularization coefficient are set to 1.0 across all models. For **ME+GD**, the retain coefficient is 1.0 for all methods. The forget coefficient is 1.0 for LLaMA 2, while it is set to 0.1 for LLaMA 2 LoRA and Phi 1.5. For the LLaMA LoRA configuration, the LoRA rank is 8 and the LoRA alpha is 32. For training LLaMA 2 with DualOptim, we use two NVIDIA H20 GPUs with 96GB of memory each. For Phi-1.5, we use two NVIDIA RTX 6000 Ada GPUs with 48GB of memory each. The detailed hyperparameters of our methods are reported in Table 12.

Table 12: Summary of hyperparameters for each method on LLM unlearning task. $\eta$ is short for learning rate.

| Model | Method | Hyperparameters |
|---|---|---|
| Phi-1.5 | ME+GD+DO | epoch $= 5$, $\eta_f = 1 \times 10^{-5}$, $\eta_r = 1 \times 10^{-5}$ |
|  | IDK+AP+DO | epoch $= 5$, $\eta_f = 8 \times 10^{-6}$, $\eta_r = 1 \times 10^{-5}$ |
| LLaMA 2 | ME+GD+DO | epoch $= 5$, $\eta_f = 1.45 \times 10^{-6}$, $\eta_r = 1 \times 10^{-5}$ |
|  | IDK+AP+DO | epoch $= 5$, $\eta_f = 1.25 \times 10^{-6}$, $\eta_r = 1 \times 10^{-5}$ |
| LLaMA 2 LoRA | ME+GD+DO | epoch $= 5$, $\eta_f = 1 \times 10^{-4}$, $\eta_r = 2 \times 10^{-4}$ |
|  | IDK+AP+DO | epoch $= 5$, $\eta_f = 5 \times 10^{-5}$, $\eta_r = 2 \times 10^{-4}$ |

# C   Additional Results

## C.1   Non-positive Correlation between Forget and Retain Gradients

Figure 4 illustrates non-positive correlation between $\widehat{g}_f$ and $\widehat{g}_r$ during unlearning, supporting the reasonableness of Assumption 3.2. Note that the results are obtained using SGD optimizer.

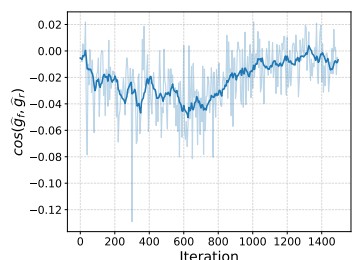

Figure 4: Cosine similarity between $\widehat{g}_f$ and $\widehat{g}_r$. The moving average curve is shown for better visualization.

## C.2   Unlearning Process of Other MU Methods

The unlearning processes of SalUn [6] and SCRUB [8] are illustrated in Figure 5 and 6, respectively. Compared to SFRon (see in Figure 2), SalUn and SCRUB exhibits less performance fluctuation. However, they underperform SFRon by a large margin. Despite that, our method is still effective in boosting their unlearning performance.

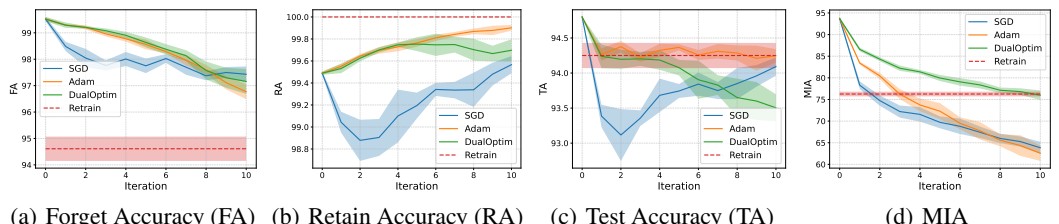

(a) Forget Accuracy (FA)   (b) Retain Accuracy (RA)   (c) Test Accuracy (TA)   (d) MIA

Figure 5: Unlearning process of SalUn. All results are obtained from unlearning 10% random subset of CIFAR-10 on ResNet-18.

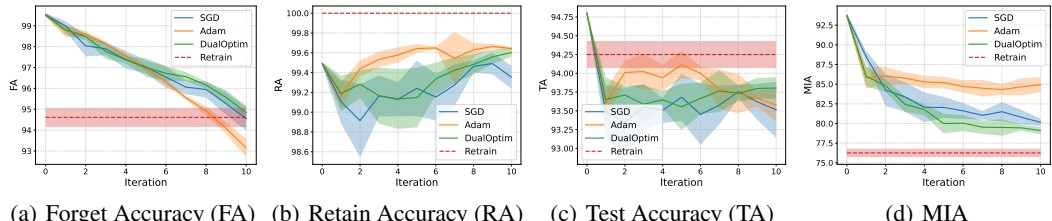

(a) Forget Accuracy (FA)   (b) Retain Accuracy (RA)   (c) Test Accuracy (TA)   (d) MIA

Figure 6: Unlearning process of SCRUB. All results are obtained from unlearning 10% random subset of CIFAR-10 on ResNet-18.

## C.3   Additional Results of Image Classification

We further conduct experiment to evaluate the performance of unlearning 50% random subsets of CIFAR-10 and 10% random subsets of CIFAR-100 and SVHN. The results are reported in Table 13. The observation in Table 13 is consistent with that in Table 1, emphasizing the efficacy of our method across different fractions of forget data, different datasets and different model architectures.

Table 13: Additional performance summary of MU methods for image classification. **(a)** 50% random subset of **CIFAR-10**, **(b)** 10% random subset of **CIFAR-100** and **(c)** 10% random subset of **SVHN**. ResNet-18 is used. All results are presented as mean and standard deviation across 5 trials with different random data. Performance gap from RT are indicated in blue.

(a) **CIFAR-10 Random Subset Unlearning (50%)**

| Method | UA | RA | TA | MIA | Gap ↓ | Std ↓ |
|---|---|---|---|---|---|---|
| Retrain | $93.36_{\pm 0.10}$ (0.00) | $100.00_{\pm 0.00}$ (0.00) | $93.11_{\pm 0.29}$ (0.00) | $69.02_{\pm 0.01}$ (0.00) | 0.00 | 0.10 |
| FT | $99.28_{\pm 0.05}$ (5.92) | $99.92_{\pm 0.03}$ (0.08) | $94.20_{\pm 0.18}$ (1.09) | $88.77_{\pm 0.28}$ (19.75) | 6.71 | 0.14 |
| GA | $95.44_{\pm 7.29}$ (2.08) | $95.53_{\pm 7.30}$ (4.47) | $90.54_{\pm 6.69}$ (2.57) | $89.14_{\pm 6.82}$ (20.12) | 7.31 | 7.03 |
| RL | $99.26_{\pm 0.09}$ (5.90) | $99.50_{\pm 0.02}$ (0.50) | $93.91_{\pm 0.12}$ (0.80) | $58.50_{\pm 1.22}$ (10.52) | 4.43 | 0.36 |
| SCRUB | $97.07_{\pm 0.28}$ (3.71) | $99.57_{\pm 0.10}$ (0.43) | $93.21_{\pm 0.17}$ (0.10) | $80.62_{\pm 1.14}$ (11.60) | 3.96 | 0.42 |
| + DualOptim | $97.42_{\pm 0.09}$ (4.06) | $99.52_{\pm 0.05}$ (0.48) | $93.38_{\pm 0.06}$ (0.27) | $77.94_{\pm 0.69}$ (8.92) | **3.43** | **0.22** |
| SalUn | $99.05_{\pm 0.09}$ (5.69) | $99.88_{\pm 0.02}$ (0.12) | $94.10_{\pm 0.15}$ (0.99) | $68.30_{\pm 0.72}$ (0.72) | 1.88 | **0.24** |
| + DualOptim | $93.82_{\pm 0.36}$ (0.46) | $97.90_{\pm 0.36}$ (2.10) | $90.66_{\pm 0.25}$ (2.45) | $68.88_{\pm 0.64}$ (0.14) | **1.29** | 0.40 |
| SFRon | $93.21_{\pm 2.34}$ (0.15) | $98.90_{\pm 0.60}$ (1.10) | $91.34_{\pm 1.22}$ (1.77) | $71.75_{\pm 3.31}$ (2.73) | 1.44 | 1.87 |
| + DualOptim | $91.38_{\pm 0.81}$ (1.98) | $99.52_{\pm 0.20}$ (0.48) | $91.07_{\pm 0.31}$ (2.04) | $69.66_{\pm 0.34}$ (0.64) | **1.29** | **0.41** |

(b) **CIFAR-100 Random Subset Unlearning (10%)**

| Method | UA | RA | TA | MIA | Gap ↓ | Std ↓ |
|---|---|---|---|---|---|---|
| Retrain | $77.96_{\pm 0.52}$ (0.00) | $99.98_{\pm 0.00}$ (0.00) | $77.23_{\pm 0.30}$ (0.00) | $41.79_{\pm 0.01}$ (0.00) | 0.00 | 0.21 |
| FT | $77.97_{\pm 0.06}$ (0.01) | $92.33_{\pm 0.54}$ (7.65) | $66.07_{\pm 0.45}$ (11.16) | $62.64_{\pm 0.18}$ (20.85) | 9.92 | 0.31 |
| GA | $81.81_{\pm 11.77}$ (3.85) | $83.52_{\pm 11.25}$ (16.46) | $62.13_{\pm 9.55}$ (15.10) | $72.62_{\pm 14.47}$ (30.83) | 16.56 | 11.76 |
| RL | $94.66_{\pm 0.21}$ (16.70) | $98.10_{\pm 0.03}$ (1.88) | $70.16_{\pm 0.09}$ (7.07) | $40.82_{\pm 0.51}$ (0.97) | 6.66 | 0.21 |
| SCRUB | $77.45_{\pm 0.68}$ (0.51) | $99.26_{\pm 0.02}$ (0.72) | $73.51_{\pm 0.37}$ (3.72) | $70.81_{\pm 0.72}$ (29.02) | 8.49 | 0.45 |
| + DualOptim | $76.72_{\pm 0.65}$ (1.24) | $99.38_{\pm 0.01}$ (0.60) | $73.87_{\pm 0.19}$ (3.36) | $63.06_{\pm 0.56}$ (21.27) | **6.62** | **0.35** |
| SalUn | $95.02_{\pm 0.40}$ (17.06) | $98.04_{\pm 0.04}$ (1.94) | $70.26_{\pm 0.29}$ (6.97) | $41.35_{\pm 0.46}$ (0.44) | 6.60 | **0.30** |
| + DualOptim | $78.04_{\pm 1.07}$ (0.08) | $97.96_{\pm 0.09}$ (2.02) | $71.10_{\pm 0.25}$ (6.13) | $41.94_{\pm 0.69}$ (0.15) | **2.10** | 0.53 |
| SFRon | $76.24_{\pm 12.35}$ (1.72) | $99.56_{\pm 0.25}$ (0.42) | $73.18_{\pm 1.30}$ (4.05) | $57.64_{\pm 7.41}$ (15.85) | 5.51 | 5.33 |
| + DualOptim | $74.32_{\pm 4.28}$ (3.64) | $99.71_{\pm 0.03}$ (0.27) | $73.66_{\pm 0.49}$ (3.57) | $54.02_{\pm 1.55}$ (12.23) | **4.93** | **1.59** |

(c) **SVHN Random Subset Unlearning (10%)**

| Method | UA | RA | TA | MIA | Gap ↓ | Std ↓ |
|---|---|---|---|---|---|---|
| Retrain | $96.05_{\pm 0.14}$ (0.00) | $99.82_{\pm 0.01}$ (0.00) | $96.53_{\pm 0.10}$ (0.00) | $79.47_{\pm 0.01}$ (0.00) | 0.00 | 0.07 |
| FT | $99.51_{\pm 0.07}$ (3.46) | $100.00_{\pm 0.00}$ (0.18) | $95.85_{\pm 0.03}$ (0.68) | $80.23_{\pm 0.66}$ (0.76) | 1.27 | 0.19 |
| GA | $96.81_{\pm 4.45}$ (0.76) | $97.41_{\pm 4.15}$ (2.41) | $92.60_{\pm 4.89}$ (3.93) | $88.80_{\pm 5.60}$ (9.33) | 4.11 | 4.77 |
| RL | $94.73_{\pm 0.32}$ (1.32) | $99.43_{\pm 0.09}$ (0.39) | $94.56_{\pm 0.24}$ (1.97) | $74.00_{\pm 1.02}$ (5.47) | 2.29 | 0.42 |
| SCRUB | $92.41_{\pm 0.25}$ (3.64) | $99.82_{\pm 0.03}$ (0.00) | $94.68_{\pm 0.11}$ (1.85) | $84.44_{\pm 0.55}$ (4.97) | 2.61 | 0.23 |
| + DualOptim | $95.42_{\pm 0.23}$ (0.63) | $99.82_{\pm 0.01}$ (0.00) | $95.06_{\pm 0.17}$ (1.47) | $82.70_{\pm 0.34}$ (3.23) | **1.33** | **0.19** |
| SalUn | $94.69_{\pm 0.22}$ (1.36) | $98.85_{\pm 0.33}$ (0.97) | $94.54_{\pm 0.31}$ (1.99) | $73.11_{\pm 0.47}$ (6.36) | 2.67 | 0.33 |
| + DualOptim | $99.58_{\pm 0.06}$ (3.53) | $100.00_{\pm 0.00}$ (0.18) | $95.76_{\pm 0.04}$ (0.77) | $79.59_{\pm 0.46}$ (0.12) | **1.15** | **0.14** |
| SFRon | $97.29_{\pm 1.98}$ (1.24) | $100.00_{\pm 0.00}$ (0.18) | $95.41_{\pm 0.36}$ (1.12) | $79.25_{\pm 3.30}$ (0.22) | **0.69** | 1.41 |
| + DualOptim | $98.20_{\pm 0.60}$ (2.15) | $100.00_{\pm 0.00}$ (0.18) | $95.63_{\pm 0.09}$ (0.90) | $79.56_{\pm 1.06}$ (0.09) | 0.83 | **0.44** |

## C.4 Additional Results of Image Generation

To further validate the effectiveness of the proposed method, we apply DualOptim to SalUn [6] on CIFAR-10 with DDPM. Table 14 illustrates that DualOptim can still enhance the performance and stability of SalUn in image generation tasks. Note that we adapt SalUn to alternate updating scheme when applying DualOptim, which originally utilizes joint updating scheme in image generation tasks. We do not include its results on ImageNet due to suboptimal and unstable performance.

Table 14: Class-wise unlearning performance of SalUn+DO on **CIFAR-10** with **DDPM**. The best unlearning performance for each forgetting class is highlighted in **bold** for FA (in %) and FID.

| Method | Automobile FA ↓ | FID ↓ | Cat FA ↓ | FID ↓ | Dog FA ↓ | FID ↓ | Horse FA ↓ | FID ↓ | Truck FA ↓ | FID ↓ | Average FA ↓ | FID ↓ |
|---|---|---|---|---|---|---|---|---|---|---|---|---|
| SalUn | 0.20 | 21.23 | 1.40 | **20.29** | 0.00 | 20.18 | 0.60 | 20.70 | 0.80 | 20.45 | $0.60_{\pm 0.49}$ | $20.57_{\pm 0.37}$ |
| +DO | **0.00** | 19.93 | **0.00** | 20.45 | **0.00** | 20.12 | **0.00** | 20.14 | **0.00** | 19.91 | $\mathbf{0.00_{\pm 0.00}}$ | $\mathbf{20.11_{\pm 0.19}}$ |

## C.5 Additional Results of Large Language Models

The results in Table 15 suggest that the alternate updating method can improve LLM unlearning. In addition, DualOptim boosts the performance based on that. As presented in Table 16, DualOptim achieves a minimal compromise in unlearning performance by using LoRA, while significantly reducing memory consumption. Furthermore, Figure 7 illustrates that the unlearning process utilizing DualOptim demonstrates greater effectiveness and stability compared to other baselines.

Table 15: Performance comparison of different updating methods on TOFU-finetuned **Phi-1.5**. The MU method is **IDK+AP**. The results include Model Capability (MC), Forget Efficacy (FE), and the average metric (Avg.) for forgetting 1%, 5%, and 10% data. Note that *Joint* represents jointly optimizing forgetting and retaining objectives, which is the default updating method; *Alternate* represents alternately optimizing these two objectives using a shared optimizer.

| Method | forget 1% data | | | forget 5% data | | | forget 10% data | | |
|---|---|---|---|---|---|---|---|---|---|
| | MC ↑ | FE ↑ | Avg. ↑ | MC ↑ | FE ↑ | Avg. ↑ | MC↑ | FE ↑ | Avg. ↑ |
| Joint | **0.4403** | 0.5723 | 0.5063 | **0.4800** | 0.5112 | 0.4956 | **0.4614** | 0.6003 | 0.5308 |
| Alternate | 0.4182 | 0.5746 | 0.4964 | 0.4348 | 0.6570 | 0.5459 | 0.4588 | 0.6619 | 0.5603 |
| **DualOptim** | 0.4221 | **0.7037** | **0.5629** | 0.4633 | **0.6974** | **0.5804** | 0.4422 | **0.7193** | **0.5807** |

Table 16: Performance comparison of different MU methods on TOFU-finetuned **LLaMA 2 with LoRA**. We set the LoRA rank to 8 and the LoRA alpha to 32. The results include Model Capability (MC), Forget Efficacy (FE), and the average metric (Avg.) for forgetting 1%, 5%, and 10% data.

| Method | forget 1% data | | | forget 5% data | | | forget 10% data | | |
|---|---|---|---|---|---|---|---|---|---|
| | MC ↑ | FE ↑ | Avg. ↑ | MC ↑ | FE ↑ | Avg. ↑ | MC↑ | FE ↑ | Avg. |
| GA+GD | 0.5007 | 0.6051 | 0.5529 | 0.5470 | 0.4306 | 0.4888 | 0.5745 | 0.9133 | 0.7439 |
| NPO+GD | 0.5290 | 0.5778 | 0.5534 | 0.5185 | 0.7032 | 0.6109 | 0.5350 | 0.7745 | 0.6548 |
| ME+GD | 0.7526 | 0.8425 | 0.7976 | **0.7435** | 0.9298 | 0.8367 | **0.7410** | 0.8856 | 0.8133 |
| **+DO** | **0.7542** | **0.9646** | **0.8594** | 0.7373 | **0.9545** | **0.8459** | 0.7363 | **0.9549** | **0.8456** |
| DPO+GD | 0.6874 | 0.7647 | 0.7260 | 0.6951 | 0.5490 | 0.6221 | 0.7308 | 0.3973 | 0.5640 |
| IDK+AP | **0.7572** | 0.6754 | 0.7163 | **0.7471** | 0.7430 | **0.7451** | **0.7604** | 0.7411 | 0.7507 |
| **+DO** | 0.7422 | **0.7729** | **0.7575** | 0.7311 | **0.7499** | 0.7406 | 0.7533 | **0.7532** | **0.7533** |

## C.6 Additional Ablation Studies

**Decouple $m$ and $v$ in Adam.** Given that Adam incorporates two momentum terms, i.e., the first moment $m$ and the second moment $v$, we conducted a further analysis to explore the impact of decoupling these terms. As shown in Table 17, decoupling both $m$ and $v$ yields the best performance. This finding reinforces the importance of fully decoupling the optimization processes for forgetting and retaining objectives.

Table 17: Ablation study on Dual Adams. Results are based on $10\%$ random subset unlearning task on CIFAR-10 using ResNet-18 pre-trained by SGD. **SFRon** is the adopted MU algorithm. $m$ and $v$ denote the **decoupled first** and **second moment terms** in Adam, respectively.

| $m$ | $v$ | FA | RA | TA | MIA | Gap ↓ | Std ↓ |
|---|---|---|---|---|---|---|---|
| ✗ | ✗ | $94.54_{\pm 2.41}$ (0.07) | $99.96_{\pm 0.02}$ (0.04) | $94.15_{\pm 0.30}$ (0.10) | $81.46_{\pm 2.42}$ (5.20) | 1.35 | 1.29 |
| ✓ | ✗ | $94.61_{\pm 2.43}$ (0.00) | $99.95_{\pm 0.02}$ (0.05) | $94.26_{\pm 0.31}$ (0.01) | $82.44_{\pm 2.56}$ (6.18) | 1.56 | 1.33 |
| ✗ | ✓ | $94.52_{\pm 2.44}$ (0.09) | $99.94_{\pm 0.03}$ (0.06) | $94.09_{\pm 0.40}$ (0.16) | $82.06_{\pm 2.94}$ (5.80) | 1.53 | 1.45 |
| ✓ | ✓ | $94.29_{\pm 1.23}$ (0.32) | $99.94_{\pm 0.01}$ (0.06) | $94.02_{\pm 0.11}$ (0.23) | $77.86_{\pm 1.39}$ (1.60) | **0.55** | **0.63** |

**Ablation studies on Adam-trained ResNet-18 and SCRUB [8].** We conduct ablation studies based on the configurations of Adam-trained ResNet-18 + SFRon and SGD-trained ResNet-18 + SCRUB. As illustrated in Table 18 and 19, Adam (F) + Adam (R) is the best configuration in both cases, indicating that the optimal optimizer for retaining is influenced by the choice of optimizer during pretraining and the specific MU algorithms employed.

**Shared optimizer with larger momentum coefficient.** We further compare DualOptim with shared optimizers that employ larger momentum coefficients. As observed in Table 20, while increasing

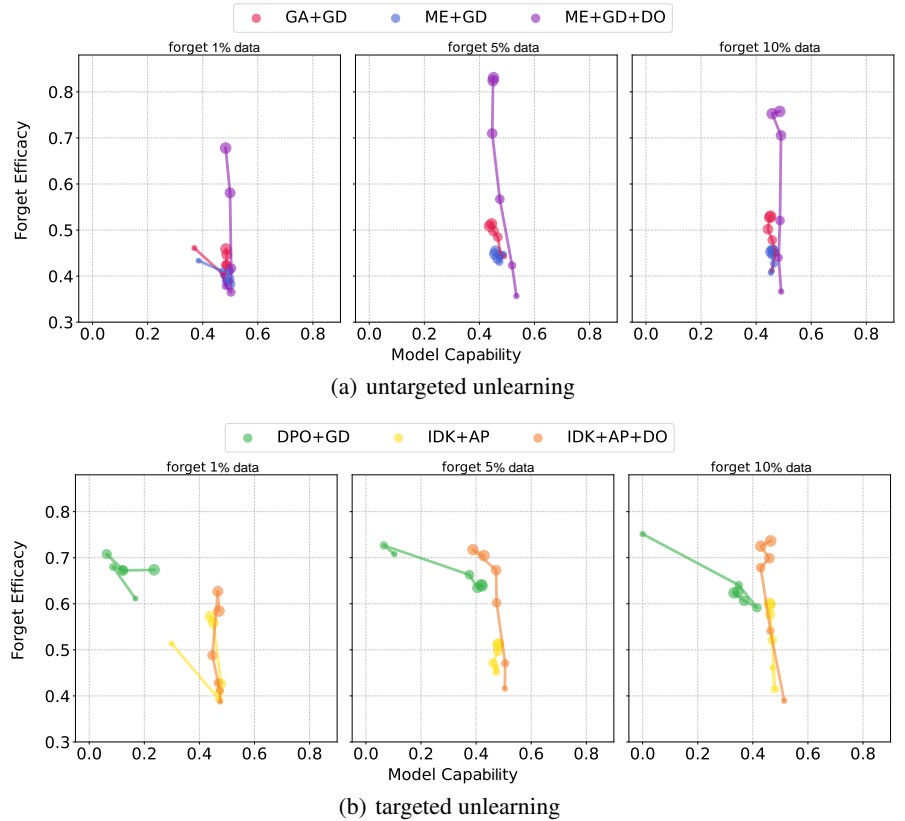

Figure 7: Forget Efficacy versus Model Capability of **(a) untargeted unlearning** and **(b) targeted unlearning** on TOFU with **Phi-1.5**. The relative size of the markers indicates the epoch of unlearning.

Table 18: Ablation study on different combinations of DualOptim. Results are based on $10\%$ random subset unlearning task on CIFAR-10 using ResNet-18 pre-trained by **Adam**. **SFRon** is the adopted MU algorithm. (F) and (R) denotes that the optimizer is used for minimizing the forget and retain losses, respectively. Note the the result of RT is reported since the pretrained model is different from that in Table 6.

| Optimizer | FA | RA | TA | MIA | Gap ↓ | Std ↓ |
|---|---|---|---|---|---|---|
| RT | $93.44_{\pm0.42}$ (0.00) | $100.00_{\pm0.00}$ (0.00) | $92.84_{\pm0.08}$ (0.00) | $78.36_{\pm0.80}$ (0.00) | 0.00 | 0.30 |
| SGD | $93.88_{\pm7.21}$ (0.44) | $98.69_{\pm2.00}$ (1.31) | $92.17_{\pm1.65}$ (0.67) | $76.91_{\pm12.23}$ (1.45) | 0.97 | 5.77 |
| Adam | $94.54_{\pm2.14}$ (1.10) | $99.61_{\pm0.11}$ (0.39) | $92.02_{\pm0.48}$ (0.82) | $76.10_{\pm4.40}$ (2.26) | 1.14 | 1.78 |
| Adam (F) + Adam (R) | $93.12_{\pm1.29}$ (0.32) | $99.33_{\pm0.03}$ (0.67) | $91.64_{\pm0.21}$ (1.20) | $78.63_{\pm0.88}$ (0.27) | **0.62** | 0.60 |
| Adam (F) + SGD (R) | $95.30_{\pm0.35}$ (1.86) | $99.45_{\pm0.09}$ (0.55) | $92.65_{\pm0.18}$ (0.19) | $78.13_{\pm0.85}$ (0.23) | 0.71 | **0.37** |

the momentum coefficient $\alpha$ can reduce performance variation when using SGD, it also results in slower convergence and ultimately leads to suboptimal performance. In contrast, increasing $\beta_1$ has only a marginal impact when using Adam. This is because Adam's momentum update rule, i.e. $\boldsymbol{m} = \beta_1 \boldsymbol{m} + (1 - \beta_1)\widehat{\boldsymbol{g}}$, inherently provides a stronger smoothing effect compared to SGD, i.e. $\boldsymbol{m} = \alpha \boldsymbol{m} + \widehat{\boldsymbol{g}}$, when the same momentum coefficient is applied.

## C.7 Visualization for Machine Unlearning in Image Generation

Generated images from the unlearned model utilizing DualOptim are shown in Figure 8 and 9. The visualization indicates that, by leveraging DualOptim, effective unlearning is achieved and the generation capability for the remaining classes is retained.

Table 19: Ablation study on different combinations of DualOptim. Results are based on $10\%$ random subset unlearning task on CIFAR-10 using ResNet-18 pre-trained by **SGD**. **SCRUB** is the adopted MU algorithm. (F) and (R) denotes that the optimizer is used for minimizing the forget and retain losses, respectively.

| Optimizer | FA | RA | TA | MIA | Gap↓ | Std↓ |
|---|---|---|---|---|---|---|
| SGD | $94.64_{\pm0.25}$ (0.03) | $99.44_{\pm0.10}$ (0.56) | $93.49_{\pm0.22}$ (0.76) | $80.37_{\pm0.86}$ (4.11) | 1.37 | 0.36 |
| Adam | $92.88_{\pm0.25}$ (1.73) | $99.62_{\pm0.10}$ (0.38) | $93.54_{\pm0.22}$ (0.71) | $82.78_{\pm0.86}$ (6.52) | 2.33 | **0.36** |
| Adam (F) + Adam (R) | $94.90_{\pm0.42}$ (0.29) | $99.52_{\pm0.09}$ (0.48) | $93.50_{\pm0.20}$ (0.75) | $78.26_{\pm0.79}$ (2.00) | **0.88** | **0.38** |
| Adam (F) + SGD (R) | $94.20_{\pm0.65}$ (0.41) | $98.68_{\pm0.17}$ (1.32) | $92.58_{\pm0.18}$ (1.67) | $78.28_{\pm1.12}$ (2.02) | 1.36 | 0.53 |

Table 20: Comparison between DualOptim and shared optimizers with larger momentum coefficients. Results are based on $10\%$ random subset unlearning task on CIFAR-10 using ResNet-18 pre-trained by **SGD**. **SFRon** is the adopted MU algorithm. Note that $\beta_1$ is the momentum coefficient in Adam .

| Optimizer | FA | RA | TA | MIA | Gap↓ | Std↓ |
|---|---|---|---|---|---|---|
| SGD ($\alpha = 0.9$) | $94.67_{\pm3.03}$ (0.06) | $99.83_{\pm0.13}$ (0.17) | $93.98_{\pm0.56}$ (0.27) | $77.80_{\pm5.61}$ (1.54) | 0.51 | 2.33 |
| SGD ($\alpha = 0.95$) | $94.44_{\pm2.14}$ (0.17) | $99.82_{\pm0.22}$ (0.18) | $94.11_{\pm0.38}$ (0.14) | $78.43_{\pm2.55}$ (2.17) | 0.67 | 1.32 |
| SGD ($\alpha = 0.99$) | $95.06_{\pm0.64}$ (0.45) | $99.02_{\pm0.09}$ (0.98) | $93.63_{\pm0.26}$ (0.62) | $78.69_{\pm1.13}$ (2.43) | 1.12 | **0.53** |
| Adam ($\beta_1 = 0.9$) | $94.54_{\pm2.41}$ (0.07) | $99.96_{\pm0.02}$ (0.04) | $94.15_{\pm0.30}$ (0.10) | $81.46_{\pm2.42}$ (5.20) | 1.35 | 1.29 |
| Adam ($\beta_1 = 0.99$) | $94.64_{\pm2.51}$ (0.03) | $99.93_{\pm0.02}$ (0.07) | $94.12_{\pm0.35}$ (0.13) | $82.10_{\pm2.40}$ (5.84) | 1.52 | 1.32 |
| Adam ($\beta_1 = 0.999$) | $94.56_{\pm2.55}$ (0.05) | $99.79_{\pm0.06}$ (0.21) | $93.75_{\pm0.38}$ (0.50) | $80.36_{\pm2.85}$ (4.10) | 1.22 | 1.46 |
| **DualOptim** | $94.69_{\pm1.13}$ (0.02) | $99.92_{\pm0.01}$ (0.08) | $94.11_{\pm0.11}$ (0.14) | $77.77_{\pm1.39}$ (1.51) | **0.44** | 0.66 |

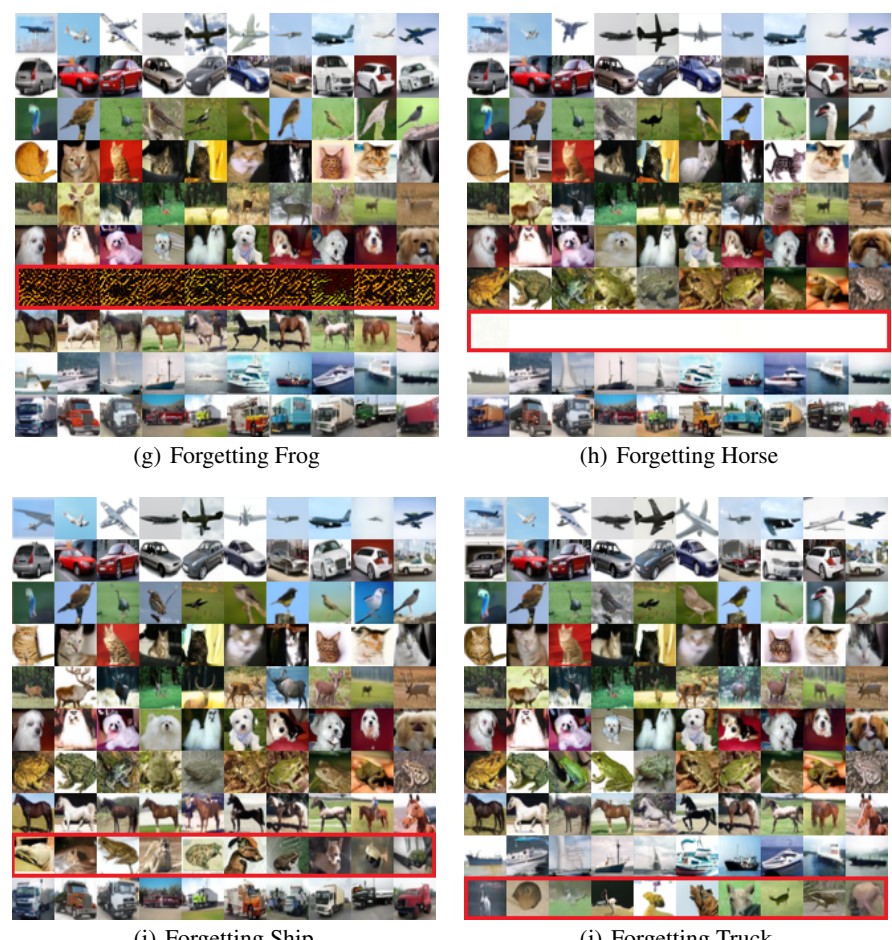

(g) Forgetting Frog

(h) Forgetting Horse

(i) Forgetting Ship

(j) Forgetting Truck

Figure 8: Visualization of class-wise unlearning results on classifier-free guidance DDPM on CIFAR-10. The forgetting class is marked with a red color.

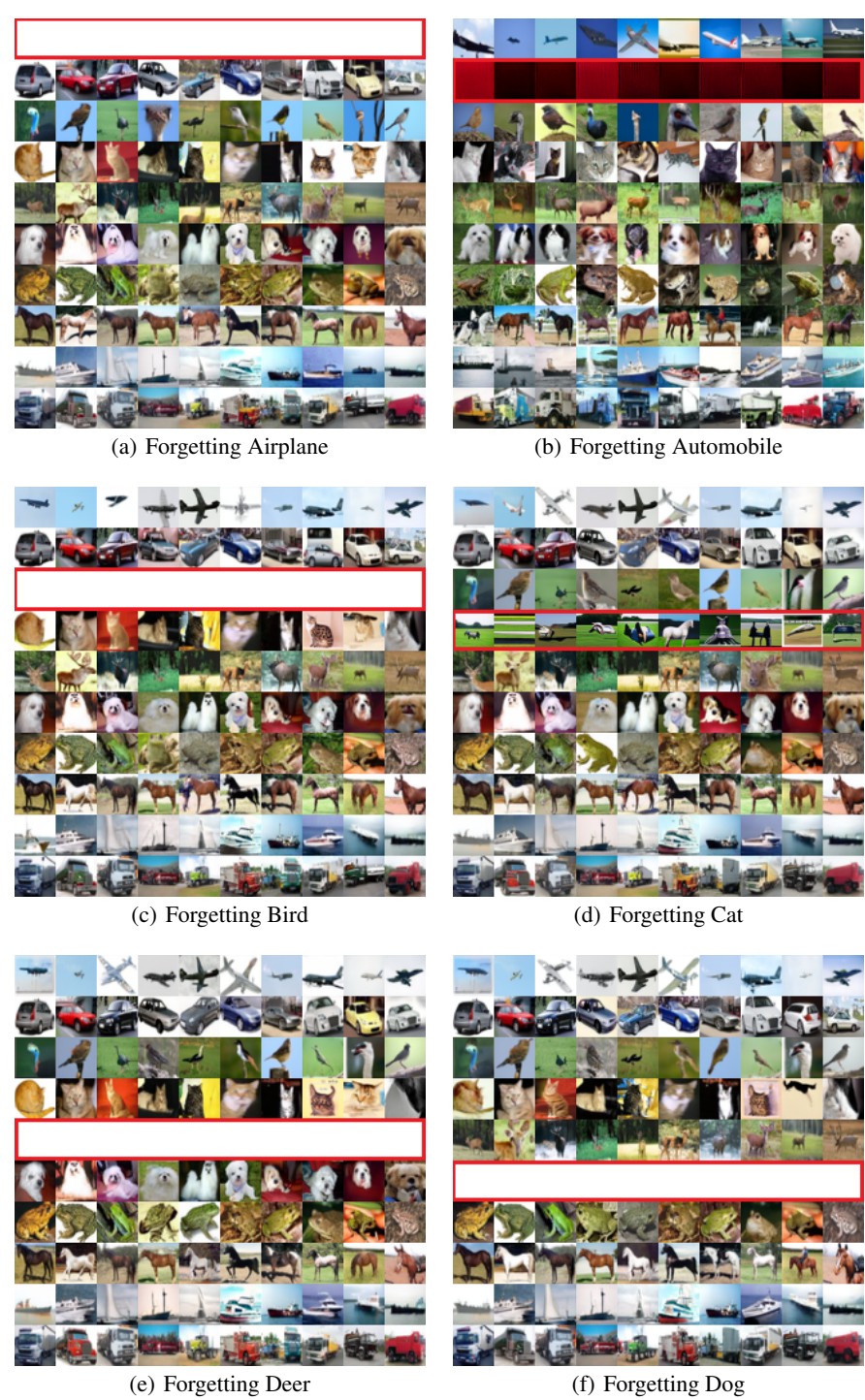

(a) Forgetting Airplane

(b) Forgetting Automobile

(c) Forgetting Bird

(d) Forgetting Cat

(e) Forgetting Deer

(f) Forgetting Dog

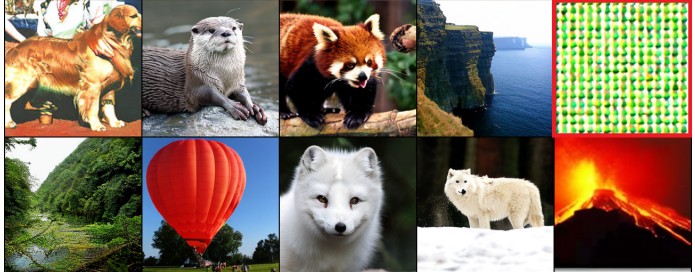

(a) Forgetting Cockatoo

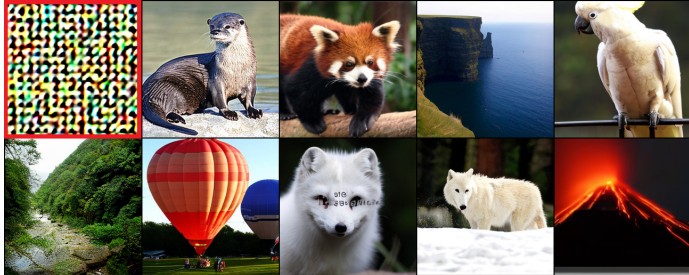

(b) Forgetting Golden Retriever

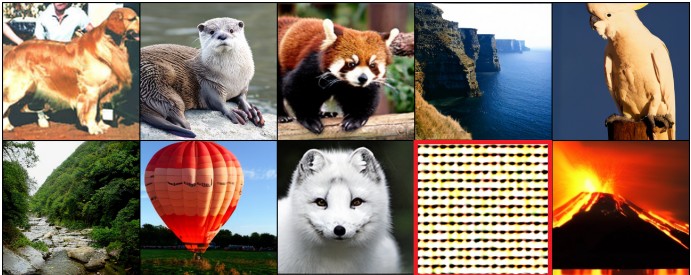

(c) Forgetting White Wolf

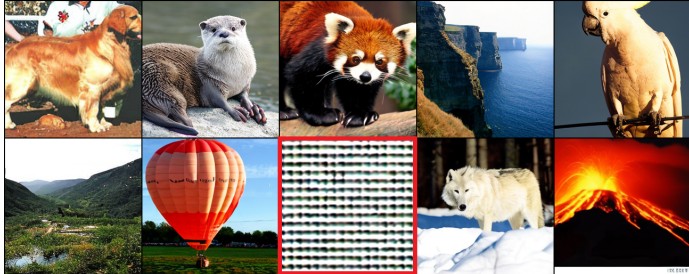

(d) Forgetting Arctic Fox

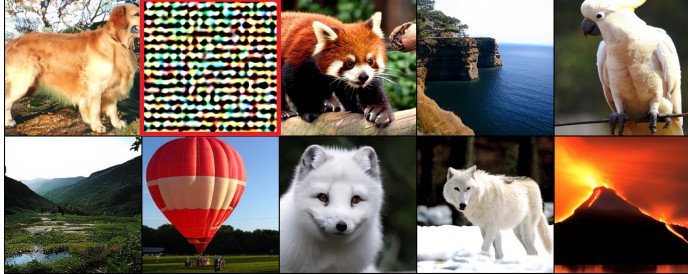

(e) Forgetting Otter

Figure 9: Visualization of class-wise unlearning results on DiT on ImageNet. The forgetting class is marked with a red color.

