# OpenReview forum: "DualOptim: Enhancing Efficacy and Stability in Machine Unlearning with Dual Optimizers"
_NeurIPS.cc/2025/Conference — NeurIPS 2025 poster_

### Official Review · Reviewer_YcdJ · 2025-06-26

**Clarity:** 2
**Significance:** 2
**Originality:** 2
**Rating:** 4
**Confidence:** 3

**Summary:**

This paper proposes a “dual-optimizer” strategy for approximate machine unlearning (MU). During unlearning, gradients from the forget loss are updated with Adam, while gradients from the retain loss continue with the model’s original optimizer (typically SGD); their momentum buffers are fully decoupled. The authors prove a tighter bound on parameter-variance under decoupled momentum and demonstrate consistent gains and markedly lower run-to-run variance across MU on image classifiers, class-wise unlearning in generative diffusion model, and 7 B-parameter LLM.

**Questions:**

- Could the authors quantify wall-clock overhead and GPU memory increase on the LLaMA-2 experiment?
- How sensitive is DualOptim when the pre-training optimizer is already adaptive (e.g., AdamW, Adan)?
- Typo in Section A.2 Proof of Theorem 3.5 Equation (11) Decoupled Momentum, $\mathbf{m}^S_{f,i}$ should be $\mathbf{m}^D_{f,i}$

**Ethical Concerns:**

["NO or VERY MINOR ethics concerns only"]

**Final Justification:**

I have read authors’ responses and discussions from all other reviewers too, I believe the authors’ responses have addressed most of my concerns. Despite the simplicity of the idea, I am satified with the comprehensive, convincing experimental results. Thus, I am happy to keep my positive rating.

**Limitations:**

Mostly yes. However, it may be beneficial to include more discussion on the conditions under which DualOptim exhibits limited performance and the reasons behind it.

**Quality:**

2

**Strengths And Weaknesses:**

Strengths
- The idea is simple yet effective, it requires minimal code change to deploy in the real world.
- The authors provides very comprehensive empirical validation with convincing details on implementation and experimental setup. Open sourcing the code and results will help the unlearning community greatly.
- The theoretical analysis provides a clear variance bound links the decoupled-momentum idea to stability, providing more than just empirical intuition.

Weaknesses
- Assumption 3.2 treats forget-retain gradient correlation as negligible, yet Figure 4’s cosine-similarity plot is not a rigorous test of that claim. Firstly, from intuition, forgetting objective and retaining objective is possibly highly negatively correlated during a whole training session (across iterative updates). Secondly, cosine similarity treats the *coordinates* of a gradient vector as “samples” and asks whether the two vectors are roughly orthogonal **within a single timestep**. The correlation in Assumption 3.2, by contrast, is about how two *random variables* (the gradient estimates produced by data sampling) co-vary **across different draws**. These are different statistical objects. Also, due to curse of dimensionality, the reported value in Figure 4 may simply reflect high dimensionality, not genuine near-zero correlation.
- The paper’s overall idea is slightly too incremental, the impact at conceptual level is limited. (but this weakness is supplemented by the comprehensiveness of the empirical result, which I as a reviewer do appreciate.)
- More advanced optimizers and their combinations could be included in the experimental/ablation study setup.

---

> ### Author Rebuttal · Authors · 2025-07-29
>
> Thanks for acknowledging that our method is simple yet effective, and the experiments are comprehensive. We provide the point-to-point responses to your concerns as follows:
>
> ---
>
> **W1: Assumption 3.2 treats forget-retain gradient correlation as negligible, yet Figure 4’s cosine-similarity plot is not a rigorous test of that claim.**
>
> **Reply:** We are sorry about the typo in line 153, we will replace $\rho(\widehat{g_{f,t_1}},\widehat{g_{r,t_2}})\simeq 0$ with $\rho(\widehat{g_{f,t_1}},\widehat{g_{r,t_2}}) \leq o(\tau) \simeq 0$ in the revision. We only assume that the correlation between forget gradient and retain gradient in different time steps is negligible compared with the correlation among forget gradients or the correlation among the retain gradients. In the proof, we only utilize the assumption $\rho(\widehat{g_{f,t_1}}, \widehat{g_{r,t_2}}) \leq 0$ for notation simplicity. This encompasses the situation that the forgetting gradient is negatively correalted with the retaining gradient. Therefore, **it does not change the conclusion of our theorem.**
>
> Moreover, to make the analysis more rigorous, we can also replace the last assumption as $\rho(\widehat{g_{f,t_1}},\widehat{g_{r,t_2}}) \leq \xi \ll \tau$, the conclusion still hold, because the term of $\overline{\mathrm{Var}}$ will be dominated by the term containing $\tau$.
>
> Thanks for pointing it out again, we will revise this part accordingly.
>
> ---
>
> **W2: The paper’s overall idea is slightly too incremental, the impact at conceptual level is limited.**
>
> **Reply:** We would like to highlight that **our method is simple but effective.** We believe that the coupled momentum factor is one of the fundamental reasons leading to the instability of MU methods. Empirical and theoretical evidences demonstrate effectiveness of using dual optimizers in stabilizing the MU process and improving the unlearning performance. Furthermore, our method is not task-specific, making it a versatile approach to empower existing MU algorithms. Therefore, we believe that our method can be broadly applied.
>
> ---
>
> **W3: More advanced optimizers and their combinations could be included.**
>
> **Reply:** In Table 5, we include the results of different combinations of Lion optimizer and SGD. In addition, we report the results of the combinations of recently popular **Muon** [1] and SGD under the same setting of Table 5 below. We can observe that **Muon also exhibits low variability, but does not achieve the optimal performance** compared with Adam, while the best combination: **Adam(F)+SGD(R)** reduces the average gap to **0.44** with an average std of **0.66**.
>
> | Optimizer | FA | RA | TA | MIA | Avg Gap | Avg Std |
> |---|---|---|---|---|---|---|
> | SGD(F) + Muon(R) | 95.74±0.40 | 99.61±0.03 | 94.04±0.09 | 84.03±0.77 | 2.37 | 0.32 |
> | Muon(F) + Muon(R) | 95.12±0.54 | 99.64±0.01 | 93.84±0.10 | 82.77±0.60 | 1.94 | 0.31 |
> | Muon(F) + SGD(R) | 94.57±1.12 | 99.93±0.01 | 94.13±0.12 | 72.46±1.01 | 1.01 | 0.57 |
>
> ---
>
> **Q1: Could the authors quantify wall-clock overhead and GPU memory?**
>
> **Reply:**  Thanks for your suggestion, we report the wall-clock time and peak memory usage of Adam and DualOptim on Llama 2 with full parameter tuning and LoRA below. We can observe that **under the configuration of full parameter tuning, the memory usage ratio is closer to 1 (Adam) : 1.5 (DualOptim).** Nonetheless, **under the LoRA configuration, the memory usage ratio is very close to 1:1** since the number of trainable parameters is significantly reduced. Additionally, the time usage ratio varies across different configurations and specific MU algorithms, but **DualOptim takes 1~1.5x more time than Adam in general**.
>
> While the memory overhead introduced by our method is not negligible for LLMs, we can adopt parameter-efficient fine-tuning (PEFT) methods, such as LoRA,  to significantly reduce the memory usage of optimizer states. As shown in Table 15, **unlearning with LoRA can also induce competitive performance compared to unlearning with full parameters**. We believe the development of PEFT will also broaden the application scenarios of our method.
>
> **Remark:** For LoRA, we used 1× H20 GPU; for full parameter tuning, we used 2× H20 GPUs. We implement DualOptim based on bitsandbytes (bnb) library, because it can be easier to maintain the optimizer states in the framework of transformers library, while Adam is implemented based on original pytorch. Since bnb has specially optimized the implementation of optimizers and LoRA significantly decreases the number of trainable parameters, **the memory usages of DualOptim is even smaller than that of Adam under the LoRA configuration.**
>
> ***Time Usage (s)***
> | Model | MU method | Adam | DualOptim | Ratio |
> |---|---|---|---|---|
> | Llama | IDK+AP |1366.47| 2034.98 | 1.49 |
> | Llama | ME+GD | 1163.95 | 1710.18 | 1.47 |
> | Llama w/ LoRA | IDK+AP | 1605.75 | 1939.18 | 1.21 |
> | Llama w/ LoRA | ME+GD | 1022.49 | 1045.55 | 1.02 |
>
> ***Memory Usage (GB/GPU)***
> | Model | MU method | Adam | DualOptim | Ratio |
> |---|---|---|---|---|
> | Llama | IDK+AP |59.84|86.52|1.45|
> | Llama | ME+GD | 59.83 | 86.40 | 1.44 |
> | Llama w/ LoRA | IDK+AP |31.44|30.62|0.97|
> | Llama w/ LoRA | ME+GD | 30.63 | 30.62 | 1.00 |
>
> ---
>
> **Q2 : How sensitive is DualOptim when the pre-training optimizer is already adaptive?**
>
> **Reply:** We included the results of the model pretriained with Adam in Table 17. The results show that when the pre-training optimizer is Adam, the best combination is Adam(F) + Adam(R), which is consistent with the clam that we use **the same optimizer as in pretraining to maintain the performance on the retain set** and use **the optimizer with adaptive learning rates to handle the large gradient variation for the forget loss**.
>
> ---
>
> **Q3 : Typo in Section A.2 Proof of Theorem 3.5 Equation (11)**
>
> **Reply:** Thanks for pointing this out. We will correct it in the revision.
>
> ---
>
> **We hope our responses can address your concerns in the review above and look forward to your feedback.**
>
> [1] Keller Jordan et al. Muon: An optimizer for hidden layers in neural networks.

---

> > ### Comment · Reviewer_YcdJ · 2025-08-01
> >
> > I have read authors’ responses and discussions from all other reviewers too, I believe the authors’ responses have addressed most of my concerns.
> > Despite the simplicity of the idea, I am satified with the comprehensive, convincing experimental results.
> > Thus, I am happy to keep my positive rating.

---

> > > ### Author Response · Authors · 2025-08-06
> > >
> > > We are happy that our responses addressed most of your concerns. We will incorporate the critical points during the rebuttal in the revised paper. Thanks again for your commitment in reviewing our work.

---

### Official Review · Reviewer_XHDw · 2025-07-03

**Clarity:** 3
**Significance:** 3
**Originality:** 2
**Rating:** 5
**Confidence:** 3

**Summary:**

This paper tackles the instability that plagues many machine-unlearning algorithms by introducing DualOptim, a two-optimizer scheme: one adaptive optimizer for the forget objective and another for the retain objective, with their momentum terms fully decoupled. The authors supply a variance-bound analysis explaining why decoupled momentum should reduce parameter drift, and they validate the method on image-classification, image-generation, and large-language-model tasks. Across these settings, DualOptim consistently narrows the gap to full retraining and lowers variance, suggesting it is a practical “plug-and-play” upgrade for existing MU pipelines.

**Questions:**

1. Are there examples where decoupling the optimizers actually hurts convergence?

2. Can you provide concrete measurements of memory usage and run-time overhead when scaling to large LLMs? How do these costs compare with single-optimizer baselines?

3. Have you evaluated DualOptim against automatic hyper-parameter scheduling techniques designed to stabilize MU?

**Ethical Concerns:**

["NO or VERY MINOR ethics concerns only"]

**Final Justification:**

This paper presents a technically sound and well-validated approach to improving the stability of machine unlearning via a dual-optimizer framework. While some aspects build on familiar ideas from multitask and bilevel optimization, the method’s targeted design for conflicting objectives in MU and its consistent performance gains justify acceptance.

**Limitations:**

Yes.

**Paper Formatting Concerns:**

No.

**Quality:**

3

**Strengths And Weaknesses:**

**Strengths**

1. DualOptim effectively addresses the hyperparameter sensitivity issue prevalent in existing MU methods, providing significantly improved stability across diverse deployment scenarios.

2. The approach demonstrates consistent performance gains across multiple domains including image classification, generation, and LLMs, showcasing its broad applicability.

3. The solution combines both theoretical foundations (adaptive learning rates and decoupled momentum) and extensive empirical validation, ensuring reliable unlearning performance.

**Weaknesses**

1. Using different optimizers or decoupled momentum for multi-objective training is familiar from multitask and bilevel learning. The paper does not clearly separate its contribution from those earlier ideas.

2. Theoretical assumptions like uncorrelated forget/retain gradients may not hold in practice, and the paper provides little guidance on when DualOptim could underperform.

3. Maintaining two optimizers doubles state storage and may slow training, yet the main text offers no quantitative overhead analysis, especially for large models.

---

> ### Author Rebuttal · Authors · 2025-07-29
>
> We really appreciate you acknowledge that our work is sufficient for acceptance. We provide point-to-point responses to your concerns as follows:
>
> ---
>
> **W1: The paper does not clearly separate its contribution from multi-task and bilevel learning**
>
> **Reply:** Thanks for pointing this out. We will add a brief discussion in our revised manuscript. From the optimisation perspective, empirical MU methods aims to optimise two objectives (1) minimize the loss on the retain set; (2) maximize the loss on the forget set. The second objective is unbounded, which is different from most multi-objective training problems which aim to optimize a single objective function that is a combination (e.g., a weighted sum) of the losses from all tasks. Therefore, MU methods are quite different from multi-objective training ones. Bilevel learning is defined by a hierarchical structure with an upper-level and a lower-level optimization problem. The most popular and competitive MU methods do not formulate the MU tasks as bilevel optimization problems.
>
> Although these three tasks share some commonalities, **our method is specifically designed to mitigate the unstability caused by two objectives with conflict in MU tasks.**
>
> ---
>
> **W2: Theoretical assumptions like uncorrelated forget/retain gradients may not hold in practice, and the paper provides little guidance on when DualOptim could underperform.**
>
> **Reply:** We are sorry about the typo in line 153, **we will replace $\rho(\widehat{g_{f,t_1}}, \widehat{g_{r,t_2}})\simeq 0$ with $\rho(\widehat{g_{f,t_1}}, \widehat{g_{r,t_2}}) \leq o(\tau) \simeq 0$ in the revision**. We only assume that the correlation between forget gradient and retain gradient in different time steps is negligible compared with the correlation among forget gradients or the correlation among the retain gradients.
> In the proof, we only utilize the assumption $\rho\(\widehat{g_{f,t_1}}, \widehat{g_{r,t_2}}\) \leq 0$ for notation simplicity. This encompasses the situation that the forgetting gradient is negatively correlated with the retaining gradient. Therefore, **it does not change the conclusion of our theorem.**
>
> Moreover, to make the analysis more rigorous, we can also replace the last assumption as $\rho(\widehat{g_{f,t_1}}, \widehat{g_{r,t_2}}) \leq \xi \ll \tau$,
> the conclusion still hold, because the term of $\overline{\mathrm{Var}}$ will be dominated by the term containing $\tau$.
> Thanks for pointing it out again, we will revise this part accordingly.
>
> As for the second point, according to Eq. 5 and Theorem 3.5, when the momentum coefficient $\alpha=0$, i.e., no momentum, DualOptim is equivalent to a single optimizer. However, modern optimizers always incorporate momentum for better optimization. Additionally, **our experiments show that our method consistently provides better performance in diverse tasks**.
>
> ---
>
> **W3: The main text offers no quantitative overhead analysis, especially for large models.**
>
> **Reply:**  Thanks for your suggestion, we report the wall-clock time and peak memory usage of Adam and DualOptim on Llama 2 with full parameter tuning and LoRA below. We can observe that **under the configuration of full parameter tuning, the memory usage ratio is closer to 1 (Adam) : 1.5 (DualOptim).** Nonetheless, **under the LoRA configuration, the memory usage ratio is very close to 1:1** since the number of trainable parameters is significantly reduced. Additionally, the time usage ratio varies across different configurations and specific MU algorithms, but **DualOptim takes 2% to 50% more time than Adam in general**.
>
> While the memory overhead introduced by our method is not negligible for LLMs, we can adopt parameter-efficient fine-tuning (PEFT) methods, such as LoRA,  to significantly reduce the memory usage of optimizer states. As shown in Table 15, **unlearning with LoRA can also induce competitive performance compared to unlearning with full parameters**. We believe the development of PEFT will also broaden the application scenarios of our method.
>
> **Remark:** For LoRA, we used 1× H20 GPU; for full parameter tuning, we used 2× H20 GPUs. We implement DualOptim based on bitsandbytes (bnb) library, because it can be easier to maintain the optimizer states in the framework of transformers library, while Adam is implemented based on original pytorch. Since bnb has specially optimized the implementation of optimizers and LoRA significantly decreases the number of trainable parameters, **the memory usages of DualOptim is even smaller than that of Adam under the LoRA configuration.**
>
> ***Time Usage (s)***
> | Model | MU method | Adam | DualOptim | Ratio |
> |---|---|---|---|---|
> | Llama | IDK+AP |1366.47| 2034.98 | 1.49 |
> | Llama | ME+GD | 1163.95 | 1710.18 | 1.47 |
> | Llama w/ LoRA | IDK+AP | 1605.75 | 1939.18 | 1.21 |
> | Llama w/ LoRA | ME+GD | 1022.49 | 1045.55 | 1.02 |
>
> ***Memory Usage (GB/GPU)***
> | Model | MU method | Adam | DualOptim | Ratio |
> |---|---|---|---|---|
> | Llama | IDK+AP |59.84|86.52|1.45|
> | Llama | ME+GD | 59.83 | 86.40 | 1.44 |
> | Llama w/ LoRA | IDK+AP |31.44|30.62|0.97|
> | Llama w/ LoRA | ME+GD | 30.63 | 30.62 | 1.00 |
>
> ---
>
> **Q1: Are there examples where decoupling the optimizers actually hurts convergence?**
>
> **Reply:** We did not observe such examples so far.
>
> ---
>
> **Q2: Can you provide concrete measurements of memory usage and run-time overhead when scaling to large LLMs? How do these costs compare with single-optimizer baselines?**
>
> **Reply:** Please see the response to W3.
>
> ---
>
> **Q3:Have you evaluated DualOptim against automatic hyper-parameter scheduling techniques designed to stabilize MU?**
>
> **Reply:** It should be noted that the hyperparameters of all evaluated methods are tuned through a **sophisticated search** (details in the responses to W2 for reviewer UZKz and W3 for reviewer wno1), so the results exhibit the optimal performance of the evaluated methods. Moreover, automatic scheduling techniques are embedded in specific MU methods [1,2] and are orthogonal to our method. Therefore, they are not included in our comparison.
>
> ---
>
> **We hope our responses can address your concerns.**
>
> [1] Chongyu Fan et al. Salun: Empowering machine unlearning via gradient-based weight saliency in both image classification
> and generation. ICLR 2024.
>
> [2] Zhehao Huang et al. Unified gradient-based machine unlearning with remain geometry enhancement. NeurIPS 2024.

---

> > ### Comment · Reviewer_XHDw · 2025-08-05
> >
> > I thank the authors for their detailed response, which addressed my questions, and I will maintain my score.

---

> > > ### Author Response · Authors · 2025-08-06
> > >
> > > We are happy that our responses addressed your concerns. We will incorporate the critical points during the rebuttal in the revised paper. Thanks again for your commitment in reviewing our work.

---

### Official Review · Reviewer_UZKz · 2025-07-03

**Clarity:** 3
**Significance:** 3
**Originality:** 3
**Rating:** 4
**Confidence:** 3

**Summary:**

This paper introduces DualOptim, a novel method designed to enhance the efficacy and stability of approximate machine unlearning (MU) methods. DualOptim addresses the instability and hyperparameter sensitivity of existing MU techniques by employing two key innovations: 1. adaptive learning rates for the forgetting objective and 2. decoupled momentum updates for the forgetting and retaining objectives. The method is validated through extensive experiments across diverse tasks, including image classification, image generation, and LLMs. Results demonstrate that DualOptim significantly improves the performance and stability of existing MU algorithms, making it a versatile and impactful solution.

**Questions:**

See Weaknesses

**Ethical Concerns:**

["NO or VERY MINOR ethics concerns only"]

**Final Justification:**

The additional experiments were helpful and have addressed my concerns. They enhance the completeness of the submission, but I consider them expected rather than transformative additions. I therefore keep the overall rating unchanged, while increasing the confidence score.

**Limitations:**

Yes

**Quality:**

3

**Strengths And Weaknesses:**

Strengths
1. The paper is technically sound, with strong theoretical foundations supporting the proposed method. The decoupling of momentum is rigorously analyzed, and its impact on variance reduction is well-justified.
2. Extensive experiments across multiple domains (image classification, image generation, and LLMs) demonstrate the generalizability of DualOptim.
3. The paper is clearly written and structured, with detailed explanations of the methodology and experimental setup.

Weaknesses
1. The use of decoupled momentum introduces additional memory and computational overhead, particularly for large-scale models like LLMs. While the authors discuss this limitation, further optimization strategies could be explored.
2. While the paper demonstrates the effectiveness of DualOptim, it lacks a detailed analysis of the sensitivity of its own hyperparameters (e.g., learning rates for forgetting and retaining objectives). Additionally, the experiments compare DualOptim with other methods using their default hyperparameters, which may not represent the optimal performance of these baselines. A fairer comparison would involve tuning all methods under the same conditions.

---

> ### Author Rebuttal · Authors · 2025-07-29
>
> Thanks for acknowledging that our paper is technically sound, well-written and the experiments are extensive. We provide point-to-point responses to your concerns as follows:
>
> ---
>
> **W1: While the authors discuss this limitation, further optimization strategies could be explored**
>
> **Reply:** For your reference, we report the wall-clock time and peak memory usage of Adam and DualOptim on Llama 2 with full parameter tuning and LoRA below. We can observe that **under the configuration of full parameter tuning, the memory usage ratio is closer to 1 (Adam) : 1.5 (DualOptim).** Nonetheless, **under the LoRA configuration, the memory usage ratio is very close to 1:1** since the number of trainable parameters is significantly reduced. Additionally, the time usage ratio varies across different configurations and specific MU algorithms, but **DualOptim takes 1~1.5x more time than Adam in general**.
>
> While the memory overhead introduced by our method is not negligible for LLMs, we can adopt parameter-efficient fine-tuning (PEFT) methods, such as LoRA,  to significantly reduce the memory usage of optimizer states. As shown in Table 15, **unlearning with LoRA can also induce competitive performance compared to unlearning with full parameters**. We believe the development of PEFT will also broaden the application scenarios of our method.
>
> **Remark:** For LoRA, we used 1× H20 GPU; for full parameter tuning, we used 2× H20 GPUs. We implement DualOptim based on bitsandbytes (bnb) library, because it can be easier to maintain the optimizer states in the framework of transformers library, while Adam is implemented based on original pytorch. Since bnb has specially optimized the implementation of optimizers and LoRA significantly decreases the number of trainable parameters, **the memory usages of DualOptim is even smaller than that of Adam under the LoRA configuration.**
>
> ***Time Usage (s)***
> | Model | MU method | Adam | DualOptim | Ratio |
> |---|---|---|---|---|
> | Llama | IDK+AP |1366.47| 2034.98 | 1.49 |
> | Llama | ME+GD | 1163.95 | 1710.18 | 1.47 |
> | Llama w/ LoRA | IDK+AP | 1605.75 | 1939.18 | 1.21 |
> | Llama w/ LoRA | ME+GD | 1022.49 | 1045.55 | 1.02 |
>
> ***Memory Usage (GB/GPU)***
> | Model | MU method | Adam | DualOptim | Ratio |
> |---|---|---|---|---|
> | Llama | IDK+AP |59.84|86.52|1.45|
> | Llama | ME+GD | 59.83 | 86.40 | 1.44 |
> | Llama w/ LoRA | IDK+AP |31.44|30.62|0.97|
> | Llama w/ LoRA | ME+GD | 30.63 | 30.62 | 1.00 |
>
> ---
>
> **W2: It lacks a detailed analysis of the sensitivity of its own hyperparameters. Additionally, the experiments compare DualOptim with other methods using their default hyperparameters, which may not represent the optimal performance of these baselines.**
>
> **Reply:**
> - Thanks for your suggestion. We report the performance of SFRon and SFRon+DualOptim with different forget learning rates below (we fix retaining learning rate to 0.01). It can be observed that **our method significantly stabalizes the unlearning process and achieves smaller gap to retrained models in a wide range of hyperparameter choices.**
> - We would like to clarify that for all baselines, we search for their optimal learning rates in a wide range and a fine granularity (all implementation details are shown in Appendix B). For SalUn and SFRon which use separate learning rates for retain and forget sets, we fix the retaining learning rate to a reasonable value and tune the forgetting learning rate. Specifically, **we start with a reasonable learning rate that induces effective unlearning, and search the optimal one at a granularity that is an order of magnitude smaller than the starting value**. For instance, if the starting value is 1e-4, the search range will be [5e-5, 6e-5, 7e-5, 8e-5, 9e-5, 1e-4, 1.1e-4, 1.2e-4, 1.3e-4, 1.4e-4, 1.5e-4]. Note that, since the reasonable starting learning rates of different methods vary a lot, we do not adopt the unified search range for all baselines. Therefore, **all baselines are carefully tuned to achieve the near-optimal performance**. We will include this detail in the revised version.
>
> ***SFRon***
> | Forget lr | 0.027 | 0.028 | 0.029 | 0.03 | 0.031 | 0.032 | 0.033 | 0.034 | 0.035 |
> | --- | --- | --- | --- | --- | --- | --- | --- | --- | --- |
> | Avg Gap | 2.56 | 2.03 | 1.49 | 0.98 | 0.51 | 0.54 | 0.52 | 0.9 | 1.19 |
> | Avg Std | 0.99 | 1.57 | 2.03 | 2.43 | 2.33 | 2.37 | 2.23 | 2.20 | 2.16 |
>
> ***SFRon+DualOptim***
> | Forget lr | 6e-5 | 7e-5 | 8e-5 | 9e-5 | 1e-4 | 1.1e-4 | 1.2e-4 | 1.3e-4 | 1.4e-4|
> | --- | --- | --- | --- | --- | --- | --- | --- | --- | --- |
> | Avg Gap | 1.83 | 1.43 | 1.04 | 0.72 | 0.44 | 0.52 | 0.52 | 0.76 | 1.19 |
> | Avg Std | 0.49 | 0.59 | 0.61 | 0.67 | 0.66 | 0.72 | 0.68 | 0.73 | 0.77 |
>
> ---
>
> **We hope our responses can address your concerns in the review above and look forward to your feedback.**

---

> > ### Comment · Reviewer_UZKz · 2025-08-05
> >
> > Thank you for your thoughtful response; it has clarified concerns. However, after reviewing the details, I believe my original score aligns well with my assessment of the work's overall contribution and impact, so I will maintain it.

---

> > > ### Author Response · Authors · 2025-08-06
> > >
> > > We are happy that our responses addressed your concerns. We will incorporate the critical points during the rebuttal in the revised paper. Thanks again for your commitment in reviewing our work.

---

### Official Review · Reviewer_vno1 · 2025-07-05

**Clarity:** 3
**Significance:** 3
**Originality:** 3
**Rating:** 5
**Confidence:** 3

**Summary:**

This paper introduces DualOptim, a method to resolve the instability and difficult tuning associated with existing machine unlearning (MU) methods. DualOptim employs two separate optimizers: an adaptive one, like Adam, for the volatile "forget" objective, and the original optimizer for the "retain" objective to preserve model performance. By decoupling the momentum between these two tasks, the method is shown to enhance the stability and efficacy of the unlearning process. Extensive experiments validate that DualOptim boosts performance across diverse applications, including image classification, image generation, and large language models.

**Questions:**

1. To quantify practical overhead, please report the wall-clock time and peak memory usage for DualOptim compared to a key baseline on a representative task.

2. To validate your stability claims, please detail the hyperparameter tuning budget for both DualOptim and the baselines to ensure a fair comparison.

3. Please address the mismatch between your theory (proven for SGD) and your algorithm (which uses Adam) by discussing the limitations and applicability of your proof.

**Ethical Concerns:**

["NO or VERY MINOR ethics concerns only"]

**Final Justification:**

The authors have clarified the methodology, which resolves my primary concern about the experimental setup.

The discussion with other reviewers and the authors' responses have provided a more comprehensive understanding of the paper's contributions and limitations.

The authors have committed to providing more detailed explanations in the final version of the paper, which will prevent potential misunderstandings.

**Limitations:**

yes

**Quality:**

3

**Strengths And Weaknesses:**

Strength:

1. The idea is clear and intuitive. Using Adam for the “forget” objective and SGD for the “retain” objective is easy to implement and generally sensible.

2. The experiments are comprehensive, spanning image classification, diffusion models and LLMs, showing the approach is not task-specific.

3. Table 5 systematically varies the two optimizers and supports the chosen pairing (Adam-F + SGD-R)

4. Authors acknowledge memory overhead in Broader-Impacts/Limitations, showing awareness of practical constraints

Weaknesses:

1. The entire proof is for plain SGD with momentum (η=1 is also hard-coded)  , but DualOptim actually uses Adam on the forget branch. Preconditioning fundamentally changes the variance dynamics, so the guarantees do not directly transfer.

2. The paper's theory culminates in Theorem 3.5, which proves a smaller worst-case parameter variance for decoupled momentum. However, it does not mathematically connect this reduced variance to better performance on downstream metrics like the "Gap" to retraining or Membership Inference Attack (MIA) success.

3. The paper states that all baselines were tuned via "sophisticated search," but does not detail if the search budget was equal for the baselines versus DualOptim, which introduces two separate learning rates. his potential imbalance undermines the fairness of the comparison and could be the source of the reported stability gains.

4. The main results compare DualOptim's alternating update schedule with baselines that sometimes use a joint optimization objective. While a fairer comparison is included in Appendix C.5, it is not featured in the main paper. This makes the primary results potentially misleading from a practical implementation standpoint.

---

> ### Author Rebuttal · Authors · 2025-07-29
>
> Thanks for acknowledging our idea is clear and the experiments are comprehensive. We provide point-to-point responses to your concerns as follows:
>
> ---
>
> **W1: The proof is only for SGD with momentum ($\eta=1$ is also hard-coded).**
>
> **Reply:**
> - We would like to clarify that the learning rate $\eta$ is just a scaling factor in our analysis, setting it to 1 only simplifies the notation and **does not lose the generality of our theorem at all**.
> - We focus on analyzing the effectiveness of decoupling itself in Sec. 3.3. **The effectiveness of preconditioning is discussed in Sec. 3.2 through numerical experiments.** Technically, it is quite challenging to rigorously bound the variance of multi-step Adam-optimized parameters. Nevertheless, our comprehensive experiments demonstrate the efficacy of the combination of preconditioning and decoupled momentum. Our ablation study demonstrate the effectiveness of using either preconditioning or decoupled momentum.
>
> ---
>
> **W2: Theorem 3.5 does not mathematically connect this reduced variance to better performance on downstream metrics**
>
> **Reply:**  Thanks for pointing this out, we would like to provide further analysis as follows:
>
> We let $M(\theta)$ to represent the performance of model with parameter $\theta$ where $M$ can be FA/RA/TA/MIA. Before we derive the variance bound of performance metric function $M(\theta)$, we make the following assumption:
>
> **Assumption A.1.** The performance metric function $M$ is Lipschitz continuous:
> $ \forall\theta_1,\theta_2, \|M(\theta_1)-M(\theta_2)\|\leq L_M\|\theta_1-\theta_2\|$,
> where $L_M$ is a non-negative constant.
>
> Following a similar logic to 3.4, we have the following corollary:
>
> **Corollary A.2.** If Assumption A.1 holds, and the parameter variance satisfies
>
> $\mathrm{Var}(\theta)\leq\overline{\mathrm{Var}}(\theta)$,  then we have $\mathrm{Var}(M(\theta))\leq L_M^2\overline{\mathrm{Var}}(\theta)$.
>
> *Proof.*
> Similar to the proof of Lemma 3.4, we have:
> $$\mathrm{Var}(M(\theta)) = \frac{1}{2}\iint p(\theta_1)p(\theta_2)\left\|M(\theta_1) - M(\theta_2)\right\|^2 d\theta_1 d\theta_2,$$
>
> where $\theta_1, \theta_2$ are independent copies of $\theta$ with probability density $p(\theta)$.
>
> By $\|M(\theta_1)-M(\theta_2)\|\leq L_M\|\theta_1-\theta_2\|$, we have:
>
> $$ \mathrm{Var}(M(\theta)) \leq \frac{L_M^2}{2}\iint p(\theta_1)p(\theta_2)\|\theta_1 - \theta_2\|^2 d\theta_1 d\theta_2.$$
>
> Since $\mathrm{Var}(\theta)\leq\overline{\mathrm{Var}}(\theta)$, we have $\frac{1}{2}\iint p(\theta_1)p(\theta_2)\|\theta_1 - \theta_2\|^2 d\theta_1 d\theta_2 \leq \overline{\mathrm{Var}}(\theta)$. Therefore, we can conclude $\mathrm{Var}(M(\theta)) \leq L_M^2 \overline{\mathrm{Var}}(\theta)$.
>
> **Discussion.** The final unlearned parameters are usually the parameter updated by retaining loss at the last iteration $T$, i.e., $\theta^D_{r,T}$ using decoupled momentum and $\theta^S_{r,T}$ using shared momentum.
> According to Theorem 3.5, we have $\overline{\mathrm{Var}}(\theta^D_{r,T}) \leq \overline{\mathrm{Var}}(\theta^S_{r,T})$. Let the upper bound of the performance variance $\overline{\mathrm{Var}}(M(\theta))=L_M^2 \overline{\mathrm{Var}}(\theta)$, we can derive $\overline{\mathrm{Var}}(M(\theta^D_{r,T})) \leq \overline{\mathrm{Var}}(M(\theta^S_{r,T}))$.
> Thus, **decoupled momentum can also induce less variability on performance metrics**.
>
> ---
>
> **W3: The paper does not detail if the search budget was equal for the baselines versus DualOptim, which introduces two separate learning rates**
>
> **Reply:** Thanks for your constructive comment. Competitive baselines such as SaLUn and SFRon also use two separate learning rates but a shared momentum factor. In such cases, we fix the retaining learning rate $\eta_r=1e-2$ when using SGD or $\eta_r = 1e-4$ when using Adam, and search for the optimal forgetting learning rate in a wide range and a fine granularity. Specifically, **we start with a reasonable learning rate that induces effective unlearning, and carefully search for the optimal value in a similar order of magnitude**. For instance, if the starting value is 1e-4, the search range will be [5e-5, 6e-5, 7e-5, 8e-5, 9e-5, 1e-4, 1.1e-4, 1.2e-4, 1.3e-4, 1.4e-4, 1.5e-4]. Note that, since the reasonable starting learning rates of different methods vary a lot, we do not adopt the unified search range for all baselines.
>
> For baselines with a unified learning rate, we follow a similar strategy as above to start with a reasonable learning rate and carefully search for the optimal one. In summary, **all baselines are carefully tuned to achieve the near-optimal performance**.
> We will include this detail in the revised version.
>
> ---
>
> **W4: The main results only compare DualOptim's alternating update schedule with baselines that sometimes use a joint optimization objective.**
>
> **Reply:** Thanks for pointing this out. In Sec. 4.3, we aim to compare our method with **unmodified baselines that were originally designed to use a joint optimization objective**. Nonetheless, we also include the results of their alternating variant for a more comprehensive comparison. The results of Table 14 of Appendix C.5 show that **DualOptim outperforms the original baselines and their alternating variants**. We will highlight this point in the main text of the revised manuscript to avoid confusion.
>
> ---
>
> **Q1: Report the wall-clock time and peak memory usage for DualOptim compared to a key baseline on a representative task**
>
> **Reply:**  Thanks for your suggestion, we report the wall-clock time and peak memory usage of Adam and DualOptim on Llama 2 with full parameter tuning and LoRA below. We can observe that **under the configuration of full parameter tuning, the memory usage ratio is closer to 1 (Adam) : 1.5 (DualOptim).** Nonetheless, **under the LoRA configuration, the memory usage ratio is very close to 1:1** since the number of trainable parameters is significantly reduced. Additionally, the time usage ratio varies across different configurations and specific MU algorithms, but **DualOptim takes 2% to 50% more time than Adam in general**.
>
> While the memory overhead introduced by our method is not negligible for LLMs, we can adopt parameter-efficient fine-tuning (PEFT) methods, such as LoRA,  to significantly reduce the memory usage of optimizer states. As shown in Table 15, **unlearning with LoRA can also induce competitive performance compared to unlearning with full parameters**. We believe the development of PEFT will also broaden the application scenarios of our method.
>
> **Remark:** For LoRA, we used 1× H20 GPU; for full parameter tuning, we used 2× H20 GPUs. We implement DualOptim based on bitsandbytes (bnb) library, because it can be easier to maintain the optimizer states in the framework of transformers library, while Adam is implemented based on original pytorch. Since bnb has specially optimized the implementation of optimizers and LoRA significantly decreases the number of trainable parameters, **the memory usages of DualOptim is even smaller than that of Adam under the LoRA configuration.**
>
> ***Time Usage (s)***
> | Model | MU method | Adam | DualOptim | Ratio |
> |---|---|---|---|---|
> | Llama | IDK+AP |1366.47| 2034.98 | 1.49 |
> | Llama | ME+GD | 1163.95 | 1710.18 | 1.47 |
> | Llama w/ LoRA | IDK+AP | 1605.75 | 1939.18 | 1.21 |
> | Llama w/ LoRA | ME+GD | 1022.49 | 1045.55 | 1.02 |
>
> ***Memory Usage (GB/GPU)***
> | Model | MU method | Adam | DualOptim | Ratio |
> |---|---|---|---|---|
> | Llama | IDK+AP |59.84|86.52|1.45|
> | Llama | ME+GD | 59.83 | 86.40 | 1.44 |
> | Llama w/ LoRA | IDK+AP |31.44|30.62|0.97|
> | Llama w/ LoRA | ME+GD | 30.63 | 30.62 | 1.00 |
>
> ---
>
> **Q2: Detail the hyperparameter tuning budget for both DualOptim and the baselines to ensure a fair comparison**
>
> **Reply:**  Please see the response to W3.
>
> ---
>
> **Q3: address the mismatch between your theory (proven for SGD) and your algorithm (which uses Adam)**
>
> **Reply:**  Please see the response to W1.
>
> ---
>
> **We hope our responses can address your concerns in the review above and look forward to your feedback.**

---

> > ### Comment · Reviewer_vno1 · 2025-08-05
> >
> > Thanks for the feedback and clarification. I believe my concerns have been addressed. I hope the final version will be more detailed and clear to avoid any potential misunderstandings. Based on this reply, along with the other reviewers' comments and the authors' feedback, I will be increasing my original rating.

---

> > > ### Author Response · Authors · 2025-08-06
> > >
> > > We are happy that our responses addressed your concerns and really appreciate that you raised the rating. It is super inspiring for us. We will incorporate the critical points during the rebuttal in the revised paper. Thanks again for your commitment in reviewing our work.

---

### Note · Authors · 2025-08-12

We appreciate the effort of AC and reviewers during the rebuttal period. We want to highlight the contributions of this work as follows:

1. We introduce Dual Optimizer (**DualOptim**) to **enhance the efficacy and stability of MU methods** by incorporating an adaptive learning rate and decoupled momentum. DualOptim can be seamlessly integrated into existing MU algorithms.
2. We provide **empirical and theoretical analyses** to demonstrate the contribution of DualOptim in improving unlearning performance and stability.
3. Comprehensive experiments across **diverse scenarios** such as image classification, image generation, and large language models validate the effectiveness of DualOptim in boosting and stabilizing the performance of MU methods.

In the rebuttal, we addressed the following main concerns of reviewers:

1. Provide further analysis on the mathematical connection between the reduced variability of parameters and better performance on downstream metrics. (**vno1**)
2.  Report the wall-clock time and peak memory usage for DualOptim. Although our method introduces memory overhead, the gap can be significantly narrowed down by introducing parameter-efficient fine-tuning techniques, such as LoRA. (**vno1**, **UZKz**, **XHDw**, **YcdJ**)
3. Clarify theoretical details and implementation settings. (**vno1**, **UZKz**, **XHDw**, **YcdJ**)
4. Conduct more ablation studies to validate the effectiveness of our method. (**UZKz**, **YcdJ**)

---

### Decision · Program_Chairs · 2025-09-17

**Decision:**

Accept (poster)

**Comment:**

This paper proposes a new method for resolving the instability of machine unlearning, by decoupling the "forget" and "retain" optimizations. Overall, reviewers found the paper easy to read and pretty convincing experimentally. The authors should make sure to include the results and edits promised in the rebuttal into the final paper.